# Decide When Ready: Stepwise Incremental Inference with Early-Exit in Spiking Neural Networks

## Abstract

Spiking Neural Networks (SNNs) are well-suited for low-power, low-latency dynamic visual perception due to their event-driven computation. However, existing SNNs rely on fixed time steps for training and inference, which leads to buffering requirements and mismatches with neuromorphic hardware, thus neglecting the potential for early recognition using partial event streams. In neuromorphic computing, ideal dynamic visual perception should be event-driven, with models continuously updating states based on incoming events and producing results as soon as confidence criteria are met. To address this, we propose the Spiking Incremental Recognition Network (SIREN), an incremental inference architecture designed to approximate this ideal paradigm. During training, the model processes event streams in fixed steps, while at inference it processes event frames step by step, updating states continuously and making dynamic decisions. SIREN integrates multiple spiking neuron types and a Spiking State-Space Model (S-SSM) to capture multiscale temporal dependencies. It also combines Causal Time Self-Attention (CTSA) with early-exit strategies for efficient termination. We evaluate our approach on three Dynamic Vision Sensor (DVS) datasets, achieving state-of-the-art performance in recognition tasks, including SL-Animals-DVS, DVS128-Gesture and the THU-EACT-50 subset, with accuracies of 93.33%, 97.92% and 100% respectively. Concurrently, we reduce the average inference steps from 16 to 9.5, with fewer synaptic operations (SOPs), demonstrating its potential for resource-constrained event-based recognition.

## 1 Introduction

To date, advances in embodied intelligence, edge computing, and adaptive perception have increased the demand for intelligent systems in resource-constrained environments (Deng et al., 2020). Despite success in visual recognition, artificial neural networks (ANNs) incur high computational and energy costs during inference, limiting deployment (Sze et al., 2017). For human action recognition (HAR), models must balance low power consumption and continuous perception with fast, reliable decisions from limited observations, making early recognition and prediction an emerging research focus (Ryoo, 2011; Hu et al., 2022; Lai et al., 2024). However, traditional frame-driven mechanisms suffer from data redundancy and response delays (Gallego et al., 2022), aggravating resource constraints. Neuromorphic computing is essentially an algorithm-hardware co-design paradigm (Yao et al., 2024) that enables event-driven processing, providing a solution (Amir et al., 2017; Davies et al., 2018) through the use of Dynamic Vision Sensors (DVS).

Neuromorphic hardware is non-von Neumann architecture hardware, often relies on synchronized clocks in digital chips like TrueNorth (Akopyan et al., 2015) and Loihi (Davies et al., 2018) for temporal consistency, leading to temporal errors and resource wastage during idle periods. In contrast, ideal neuromorphic system directly processes event streams from event cameras, where only a portion of spiking neurons are activated to execute sparse synaptic accumulation, enabling event-driven processing without stacking events into frames. With early-exit, the system halts computation when sufficient confidence is reached, further saving resources (Wolczyk et al., 2021). Spiking Neural Networks (SNNs), due to their spike-driven nature, align perfectly with the asynchronous, sparse processing characteristics of neuromorphic hardware.

Although SNNs are promising (Karamimanesh et al., 2025), their inference relies on discrete time steps. At each time step, the event stream is accumulated and propagated (Luo et al., 2024; Cai et al., 2024; Lin et al., 2024). This framework imposes a synchronization constraint. Despite the asynchronous and sparse nature of input events, the network still waits for the clock to trigger propagation and accumulated inputs at each predefined step (Du et al., 2025). This results in a pseudo event-driven system that departs from fully event-driven computing (Marostica et al., 2025).

Motivated by these considerations, we propose the Spiking Incremental Recognition Network (SIREN). Inspired by (Zubic et al., 2024), we extend the Spikformer (Zhou et al., 2023) by adding the Spiking State-Space Model (S-SSM) structure and introduce a causal time self-attention mechanism at the macro level. At the micro level, to address the challenge of incremental inference, we add causal masking to the time self-attention layers, enabling the network to process event frames stepwise at inference while training with full time steps. To improve multiscale temporal memory, we use Leaky Integrate-and-Fire (LIF) and Resonate-and-Fire (RF) neurons. To realize the above incremental inference behavior, we require an inference exit mechanism. We implement a smooth exit mechanism with a patience parameter, enabling early-exit based on the entropy of confidence distribution. We evaluate SIREN on three DVS human action recognition datasets and demonstrate its advantages over state-of-the-art methods as shown in Fig. 1(b). Our contributions are summarized as follows:

1. **Incremental inference framework.** We systematically introduce an incremental inference framework, moving towards more hardware-friendly dynamic inference.

2. **SIREN architecture.** We introduce SIREN, an incremental inference architecture that leverages spiking state-space modeling, causal time self-attention, and an entropy-based early-exit strategy to achieve efficient and accurate dynamic inference.

3. **Performance.** We evaluate SIREN on SL-Animals-DVS, achieving 93.33% accuracy, which is approximately 2% higher than the state-of-the-art, reducing the average inference length to 9.5 of 16 steps. This demonstrates lower computational redundancy and highlights its suitability for low-power edge devices.

4. **Power efficiency.** We evaluate the energy consumption in terms of SOPs on the SL-Animals-DVS sign language recognition dataset. With the early-exit mechanism, we achieved a theoretically low number of synaptic operations (SOPs), providing an analytical indication of energy efficiency.

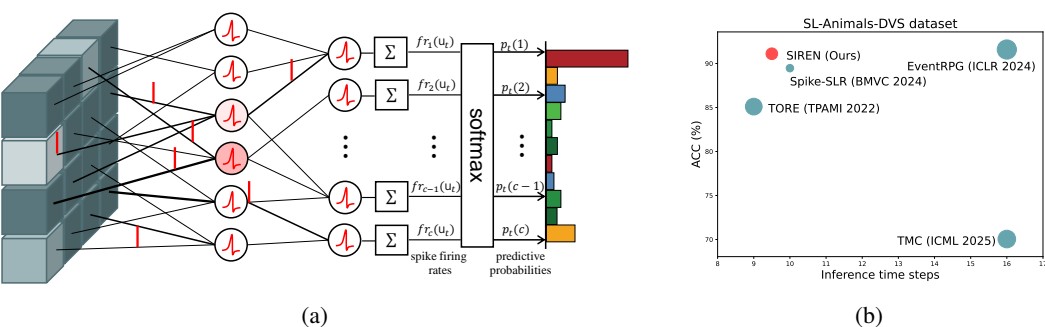

(a)           (b)

Figure 1: (a) DVS-SNN Cooperative Model for $C$-class Classification: Events from the DVS are fed directly to their position-matched spiking neurons. Output layer neurons process the input $u_t$, generating spikes and producing firing rates $fr(u_t)$ for $C$ classes. These rates are then passed through a softmax function to yield classification probabilities $p_t(c)$. (b) Accuracy vs inference time steps, with marker size representing the model's parameter count.

## 2 RELATED WORK

Relevant prior work includes studies of dynamic inference strategies, temporal generalization in training, and multiscale temporal modeling, all aiming to relax the reliance on fixed time steps.

**Dynamic Inference Strategies:** Dynamic inference strategies, extensively studied in ANNs, selectively activate different parts of the model at runtime. In SNNs, dynamic inference focuses on adjusting inference time based on the input data. Early works (Li et al., 2023a) introduced the Dynamic Confidence mechanism, along with early exit in SNNs (Li et al., 2023b), enabling early termination when sufficient confidence is reached, spurring numerous follow-up studies on time step adjustments and early-exit mechanisms (Chen et al., 2025; Ding et al., 2024; Du et al., 2025). Chen et al. (2025) proposed delay-adaptive classifiers with conformal prediction for reliability, while Ding et al. (2024) explored dynamic temporal resource allocation. Wu et al. (2025) and Zhou et al. (2024) integrated cutoff mechanisms, including top-k selection and regularization, to terminate inference early. In addition, the Hybrid Step-Wise Distillation SNNs (Zhong et al., 2024) explicitly target latency reduction by encouraging consistent step-wise predictions, representing an important line of work on inference-time temporal adaptivity. These studies highlight the broader potential of temporal adaptivity in SNNs, yet they often lack robust memory architectures, rely on brittle confidence estimators, and rarely evaluate on real DVS datasets.

**Temporal Generalization in Training:** Temporal generalization strategies aim to enhance the robustness of SNNs across varying time steps during training, reducing reliance on fixed steps. Du et al. (2025) applied temporal resampling to improve network performance across different time scales, while Luo et al. (2024) proposed integer-valued training with spike-driven inference to achieve stable temporal representations. Shan et al. (2025) introduced a spiking multiscale attention module with Attention ZoneOut like Yin et al. (2024) to capture multiscale spatio-temporal interactions, and Srinivasan & Roy (2021) developed block-wise conditional training and inference. Beyond these, adaptive time-step training has also been explored for spike-based neural rendering, such as NeRFs (Lin et al., 2025), which automatically explores the trade-off between rendering quality and time-step length during training and enables scene-adaptive inference with variable time steps. Despite their effectiveness in improving temporal robustness, these methods do not provide inference-time incremental processing and still face limitations in computational efficiency (Cao et al., 2024b).

**Multiscale Temporal Architecture Design:** Multiscale Temporal modeling includes both micro-level and macro-level approaches. At the micro level, Zheng et al. (2024) introduced temporal dendritic heterogeneity, where neurons are equipped with multiple dendritic branches. At the macro level, Shan et al. (2025) proposed a spiking multiscale attention module to capture spatio-temporal interactions, while Tan et al. (2024) applied multi-granularity frame partitioning for event-based lip-reading.

## 3 PRELIMINARIES

We recall the linear dynamics of LIF and RF neurons without reset, formulating them as ordinary differential equations (ODEs) that admit closed-form causal solutions

$$\frac{d}{dt}V(t) = -V(t) + I(t), \qquad V(t) = e^{-(t-t_0)}V(t_0) + \int_{t_0}^{t} e^{-(t-\tau)}I(\tau)\,d\tau \tag{1}$$

$$\frac{d}{dt}z(t) = (-b+i\omega)\,z(t) + I(t), \qquad z(t) = e^{(-b+i\omega)(t-t_0)}z(t_0) + \int_{t_0}^{t} e^{(-b+i\omega)(t-\tau)}I(\tau)\,d\tau \tag{2}$$

where $I(t)$ is the input current, $V(t)$ is the LIF membrane potential, $z(t) \in \mathbb{C}$ is the RF state, $b > 0$ is the decay, $\omega \in \mathbb{R}$ is the resonance frequency, $t_0$ is the initial time. Both models are causal weighted integrators (convolutions with kernels $e^{-(\cdot)}$ and $e^{(-b+i\omega)(\cdot)}$). The LIF neuron acts as a low-pass filter, while the RF neuron adds an oscillatory component that yields resonant behavior.

## 4 METHODOLOGY

This section details the SIREN architecture, the Spiking State-Space Model (S-SSM), the Causal Spatial-Temporal Self-Attention (C-STSA) mechanism, and the incremental inference procedure.

## 4.1 OVERALL ARCHITECTURE

Fig. 2 presents the overall architecture of SIREN and its distinct processing strategies during training and inference. At the core of SIREN is the ChronoSpikFormer backbone, a spiking Transformer structure composed of three key parts for feature extraction, spatio-temporal attention, and state memory. During training, events are first aggregated into fixed-length frame sequences and then processed by ChronoSpikFormer, while during inference, event frames arrive sequentially, incrementally update the state, and an early-exit mechanism decides when to stop inference.

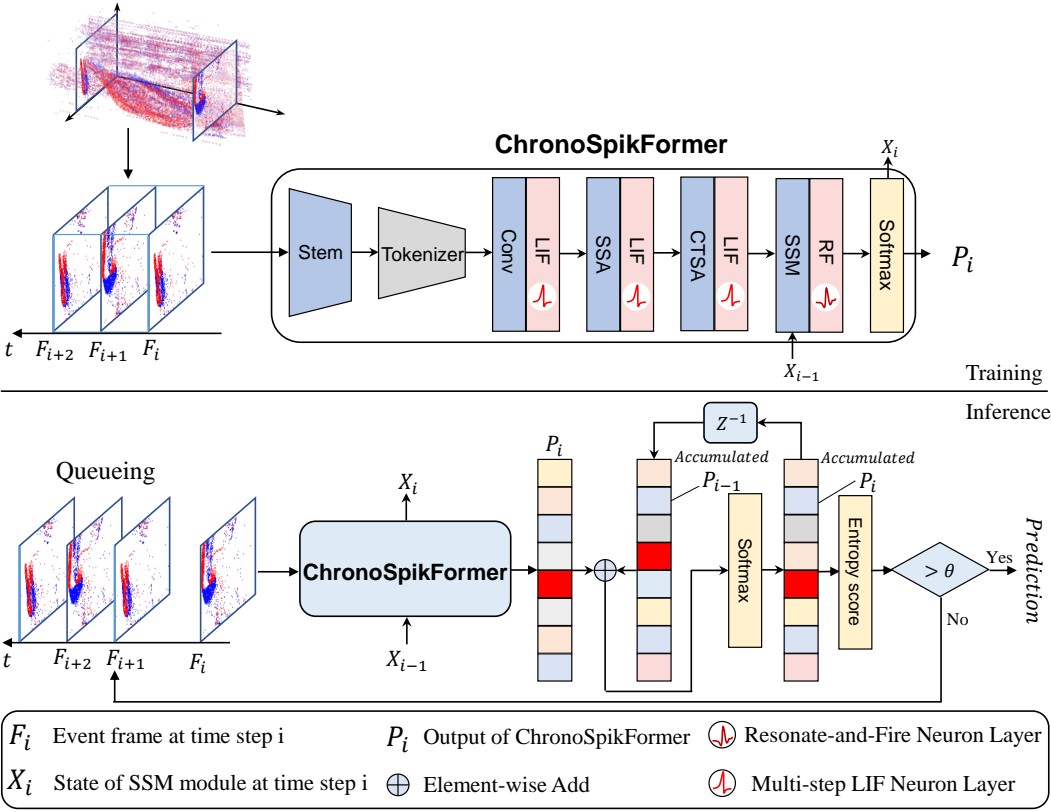

Figure 2: SIREN framework in training and inference modes. The ChronoSpikFormer backbone consists of a stem and tokenizer implemented with convolutional operations, followed by the spatial self-attention (SSA) module, the causal time self-attention (CTSA) module, and the spiking state-space model (S-SSM) module consisting of SSM and RF spiking technique. During training, SIREN operates with fixed time steps. During inference, event frames $F_i$ arrive sequentially and incrementally update the S-SSM state $X_i$, producing class probabilities $P_i$. The entropy score of $P_i$ is monitored at each step, and once it exceeds a predefined threshold, SIREN performs early exit at time step $i$.

The incremental inference of SIREN relies on two key aspects. First, the ability to integrate event streams using spiking neurons and a spiking state-space model (S-SSM) that updates temporal states. Second, an entropy-based early-exit strategy enables dynamic inference steps, reducing redundant computation. To support temporal integration, SIREN uses a hierarchical memory mechanism that ensures both dynamic responsiveness and stable early-exit decision-making. This mechanism integrates LIF neurons, RF neurons, and the S-SSM.

## 4.2 SPIKING STATE-SPACE MODEL

As shown in Section 3, LIF and RF neurons, as first-order systems with fixed decay dynamics, are limited in modeling complex temporal patterns. To capture long-range or non-stationary dependen-

cies, we use a spiking state-space module (S-SSM).

$$\frac{d}{dt}\boldsymbol{x}(t) = \boldsymbol{A}\,\boldsymbol{x}(t) + \boldsymbol{B}\,\boldsymbol{u}(t), \qquad \boldsymbol{y}(t) = \boldsymbol{C}\,\boldsymbol{x}(t) + \boldsymbol{D}\,\boldsymbol{u}(t) \tag{3}$$

where input $\boldsymbol{u}(t) \in \mathbb{C}^M$, state vector $\boldsymbol{x}(t) \in \mathbb{C}^H$, and output $\boldsymbol{y}(t) \in \mathbb{C}^N$. The state matrix $\boldsymbol{A} \in \mathbb{C}^{H \times H}$ governs memory dynamics, while $\boldsymbol{B}$, $\boldsymbol{C}$, and $\boldsymbol{D}$ denote the input matrix, readout matrix, and skip matrix. If $\boldsymbol{A}$ is diagonalizable we write $\boldsymbol{A} = \boldsymbol{V}\,\boldsymbol{\Lambda}\,\boldsymbol{V}^{-1}$ and work in the modal basis $\tilde{\boldsymbol{x}}(t) = \boldsymbol{V}^{-1}\boldsymbol{x}(t)$

$$\frac{d}{dt}\tilde{\boldsymbol{x}}(t) = \boldsymbol{\Lambda}\,\tilde{\boldsymbol{x}}(t) + \tilde{\boldsymbol{B}}\,\boldsymbol{u}(t), \qquad \boldsymbol{y}(t) = \tilde{\boldsymbol{C}}\,\tilde{\boldsymbol{x}}(t) + \boldsymbol{D}\,\boldsymbol{u}(t) \tag{4}$$

with $\boldsymbol{\Lambda} = \mathrm{diag}(\lambda_1, \ldots, \lambda_H), \tilde{\boldsymbol{B}} = \boldsymbol{V}^{-1}\boldsymbol{B}, \tilde{\boldsymbol{C}} = \boldsymbol{C}\boldsymbol{V}$. We do not recompute an eigendecomposition during training. Instead, we directly learn the diagonal modal dynamics via

$$\lambda_i = -\exp(\rho_i) + i\,\omega_i, \quad \rho_i, \omega_i \in \mathbb{R} \tag{5}$$

this enforces the stability condition $\Re(\lambda_i) < 0$. The change-of-basis matrix $\boldsymbol{V}$, determined by the SSM construction, serves as a fixed transformation.

Accordingly, training implicitly learns $\boldsymbol{A} = \boldsymbol{V}\boldsymbol{\Lambda}\boldsymbol{V}^{-1}$ through the parameters $\rho_i$ and $\omega_i$. To align with implementation, we fold the input projection into the preceding layer via $\tilde{\boldsymbol{u}}(t) = \tilde{\boldsymbol{B}}\,\boldsymbol{u}(t)$, yielding

$$\frac{d}{dt}\tilde{\boldsymbol{x}}(t) = \boldsymbol{\Lambda}\,\tilde{\boldsymbol{x}}(t) + \tilde{\boldsymbol{u}}(t) \tag{6}$$

We apply zero-order hold (ZOH) discretization. With sampling step $\Delta \in \mathbb{R}$ ($\Delta > 0$) and piecewise-constant input $\tilde{\boldsymbol{u}}(t) = \tilde{\boldsymbol{u}}_k$ for $t \in [k\Delta, (k+1)\Delta)$, the exact ZOH discretization is

$$\tilde{\boldsymbol{x}}_{k+1} = \bar{\boldsymbol{N}}\,\tilde{x}_k + \bar{\boldsymbol{S}}\,\tilde{u}_k, \tag{7}$$

$$\bar{\boldsymbol{N}} = e^{\Delta\boldsymbol{\Lambda}}, \;\; \bar{\boldsymbol{S}} = \boldsymbol{\Lambda}^{-1}\big(e^{\Delta\boldsymbol{\Lambda}} - \boldsymbol{I}\big) \tag{8}$$

which decouples per mode

$$\tilde{x}_{i,k+1} = \bar{n}_i\,\tilde{x}_{i,k} + \bar{s}_i\,\tilde{u}_{i,k}, \tag{9}$$

$$\bar{n}_i = e^{\Delta\lambda_i}, \quad \bar{s}_i = \frac{e^{\Delta\lambda_i} - 1}{\lambda_i} \tag{10}$$

we define $\bar{s}_i$ as $\bar{s}_i = (\exp(\Delta_i\lambda_i) - 1)/\lambda_i$ to avoid cancellation. For a sufficiently small number $\varepsilon$, when $|\Delta\lambda_i| < \varepsilon$ we switch to the limit $\bar{s}_i \approx \Delta$. We also allow a learnable per-mode step $\Delta_i = \gamma\,e^{\eta_i}$ (global scale $\gamma > 0$). We omit the skip matrix $\boldsymbol{D}$ and perform spiking after mapping back to the original basis

$$\boldsymbol{s}_k = \Theta\big(\boldsymbol{V}\,\tilde{\boldsymbol{x}}_k\big), \qquad \boldsymbol{y}_k = \boldsymbol{C}\,\boldsymbol{s}_k \tag{11}$$

where $\Theta(\cdot)$ is the Heaviside step function, $\boldsymbol{s}_k$ is the spike readout at time $k$, $\boldsymbol{y}_k$ is the final output. In deep stacks the readout of layer $\ell$ and the input projection of layer $\ell+1$ compose into a single learnable map. Keeping a separate $\boldsymbol{D}$ bypasses the spike nonlinearity and causes input leakage without adding expressivity, so we absorb $\boldsymbol{C}$ into the next layer and set $\boldsymbol{D} = \boldsymbol{0}$. Although $\tilde{\boldsymbol{x}}_k$ can be complex when $\omega_i \neq 0$, we ensure $\boldsymbol{s}_k \in \mathbb{R}^N$ by (i) using conjugate-symmetric modal pairs and (ii) applying a real spiking head $\Theta(\cdot)$ to $\boldsymbol{V}\tilde{\boldsymbol{x}}_k$.

Stability follows from $\rho(\bar{\boldsymbol{N}}) = \max_i |e^{\Delta\lambda_i}| = e^{\Delta\,\Re(\lambda_i)} < 1$ since $\Re(\lambda_i) = -e^{\rho_i} < 0$. Implementation details for the ZOH derivation, the real $2 \times 2$ equivalent of complex modes, per-mode step learning, and additional numerical safeguards are provided in Appendix A.1.1.

## 4.3 Causal Spatial-Temporal Self-Attention

We propose a Causal Spatial-Temporal Self-Attention (C-STSA) mechanism, inspired by divided attention strategies, to efficiently capture spatio-temporal dependencies in event streams. C-STSA employs softmax-free Causal Time Self-Attention (CTSA), as shown in Fig. 3, enhancing temporal consistency in offline training while enabling online inference through a causal time mask.

For per-step intermediate features $\boldsymbol{X}_t$, we define spike-generated queries, keys and values as

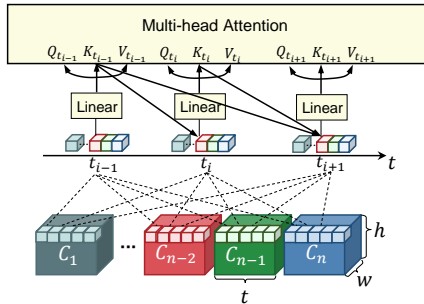

$$\boldsymbol{Q}_t = \mathcal{S}_Q(\boldsymbol{X}_t),$$
$$\boldsymbol{K}_{t'} = \mathcal{S}_K(\boldsymbol{X}_{t'}), \quad \boldsymbol{V}_{t'} = \mathcal{S}_V(\boldsymbol{X}_{t'}) \tag{12}$$

split into $h$ heads of width $d_h$ ($C = h\,d_h$). With a causal mask $\boldsymbol{M}_{t,t'} = \mathbf{1}(t' \leq t)$, scores and aggregation are

$$\beta_{t,t'} = \sigma\,\boldsymbol{Q}_t\,\boldsymbol{K}_{t'}^{\top}\,\boldsymbol{M}_{t,t'}, \qquad \sigma = \frac{\zeta_h}{\sqrt{d_h}}, \ \ \zeta_h > 0 \tag{13}$$

$$\mathrm{CTSA}_t = \Theta\Big(\sum_{t' \leq t} \beta_{t,t'}\,\boldsymbol{V}_{t'}\Big) \tag{14}$$

Figure 3: Causal Time Self-Attention (CTSA) for $n$-channel data. Different colors denote different channels.

where $\Theta(\cdot)$ denotes the spiking activation, and $\sigma = \zeta_h/\sqrt{d_h}$ with $\zeta_h > 0$ a learnable temperature parameter initialized to 1. Heads are linearly recombined after aggregation.

A softmax layer isn't required here because the following properties jointly ensure stability, well-conditioned scaling, and sparsity without exponential normalization. (i) *Bounded scores:* With unit-norm features or bounded spikes, the attention scores satisfy $|\beta_{t,t'}| \leq \sigma$ within the active support, ensuring that their magnitudes do not grow with $d_h$ (where $\sigma = \zeta_h/\sqrt{d_h}$). (ii) *Length stability:* Applying optional length normalization $w_{t,t'} = \beta_{t,t'}/\max(1, Z_t)$ with $Z_t = \sum_{t' \leq t} \boldsymbol{M}_{t,t'}$ ensures that $\|\sum_{t' \leq t} w_{t,t'} \boldsymbol{V}_{t'}\|$ remains insensitive to the history length. (iii) *Lipschitz control:* The resulting linear operator has spectral norm bounded by $\sigma$, and with 1-Lipschitz spiking surrogates, both forward activations and gradients stay well-conditioned. (iv) *Sparsity:* Since $\boldsymbol{Q}_t$ and $\boldsymbol{K}_{t'}$ are spike-sparse and $\boldsymbol{M}$ is causal, only active past steps contribute, and the expected number of temporal links scales as $r_Q r_K Z_t$ with firing rates $r_Q, r_K$, reducing complexity from $\mathcal{O}(Z_t d_h)$ to $\mathcal{O}(r_Q r_K Z_t d_h)$. Further details, including proof sketches, choices between unit-norm and bounded spikes, and ablations, are provided in Appendix A.1.2.

### 4.4 INCREMENTAL INFERENCE

To efficiently process real-time event streams from DVS, we use an entropy-based early-exit mechanism, where the entropy of the confidence distribution serves as the stopping criterion.

We use the entropy score $s_t = 1 - H(\widehat{\boldsymbol{p}}_t)/\log C \in [0, 1]$, where higher $s_t$ means lower entropy (more confident). We stop when $s_t \geq \theta$ for $\kappa$ consecutive steps and the predicted label is stable. The stopping rule is as follows.

The early-exit decision rule with patience is based on entropy calculation. The optimal threshold $\theta$ is determined using the Pareto frontier. To select an appropriate scoring metric, we evaluate three candidates based on the area under the curve (AUC): (i) predictive entropy, (ii) maximum confidence, and (iii) the confidence margin between the top-1 and top-2 predictions. The entropy score was found to be the most appropriate, as detailed in Appendix A.1.3.

At time step $t$, the readout produces class evidence $\boldsymbol{z}_t \in \mathbb{R}^C$. We form temperature-scaled probabilities:

$$\boldsymbol{p}_t = \mathrm{softmax}\big(\boldsymbol{z}_t/T\big), \qquad p_t(c) = \frac{\exp(z_t(c)/T)}{\sum_{c'} \exp(z_t(c')/T)} \tag{15}$$

where, $c \in 1, ..., C$ denotes the class index, and $\boldsymbol{p}_t$ the class probability vector with component $p_t(c)$. The predicted class is $\widehat{y}_t = \arg\max_c \widehat{p}_t(c)$. We define the entropy score $s_t^{\mathrm{ent}}$ as:

$$s_t^{\mathrm{ent}} = 1 - \frac{H(\widehat{\boldsymbol{p}}_t)}{\log C}, \qquad H(\widehat{\boldsymbol{p}}_t) = -\sum_{c=1}^{C} \widehat{p}_t(c)\,\log\widehat{p}_t(c) \tag{16}$$

Algorithm 1 details the inference process for an input event stream, where the event stream is split into $L$ event frames, and an exponential moving average with coefficient $\alpha$ is used to smooth the confidence scores, suppressing transient spikes while introducing only a small temporal lag.

---

**Algorithm 1:** Incremental Inference with Entropy-based Early-Exit in SIREN

---

**Input:** $U = \{u_1, \ldots, u_L\}$
**Parameter:** threshold $\theta$, patience $\kappa$, EMA coefficient $\alpha \in [0, 1)$, temperature $T > 0$.
**Output:** Predicted label $\widehat{y}$

1   Initialization: $x \leftarrow 0$; $step \leftarrow 0$; $\widehat{y}_{\text{last}} \leftarrow \text{None}$;
2   $(z_1, x) \leftarrow \text{ChronoSpikFormer}(u_1, x)$; $p_1 \leftarrow \text{softmax}(z_1/T)$; $\widehat{p}_1 \leftarrow p_1$; $\widehat{y}_1 \leftarrow \arg\max_c \widehat{p}_1(c)$;
3   **for** $t \leftarrow 2$ **to** $L$ **do**
4      $(z_t, x) \leftarrow \text{ChronoSpikFormer}(u_t, x)$;    *// one-step update (ChronoSpikFormer shown in Fig. 2)*
5      $\widehat{p}_t \leftarrow \alpha \widehat{p}_{t-1} + (1 - \alpha) \text{softmax}(z_t/T)$;    *// Use Exponential Moving Average*
6      $\widehat{y}_t \leftarrow \arg\max_c \widehat{p}_t(c)$;
7      $H_t \leftarrow -\sum_{c=1}^{C} \widehat{p}_t(c) \log \widehat{p}_t(c)$; $score_t \leftarrow 1 - H_t / \log C$;    *// Calculate entropy score by Eq. 16*
8      $stable \leftarrow (\widehat{y}_{\text{last}} = \text{None}) \vee (\widehat{y}_t = \widehat{y}_{\text{last}})$;
9      **if** $score_t \geq \theta \wedge stable$ **then** $step \leftarrow step + 1$;
10      **else** $step \leftarrow 0$;
11      $\widehat{y}_{\text{last}} \leftarrow \widehat{y}_t$;    *// Stability-patience rule*
12      **if** $step \geq \kappa$ **then**
13         **return** $\widehat{y}_t$.

14   **return** $\widehat{y}_L$.

---

# 5 EXPERIMENTS

In this section, we evaluate the effectiveness of SIREN. All experiments are implemented in PyTorch and conducted on a single NVIDIA RTX 4090 GPU. We assess performance on three DVS datasets: SL-Animals-DVS (Vasudevan et al., 2022), DVS128-Gesture (Amir et al., 2017), and THU-EACT-50 (Gao et al., 2023), focusing on accuracy under fixed time steps, latency-aware accuracy at each inference step, and a preliminary energy analysis. Data preprocessing and training settings are detailed in Appendix A.2.

## 5.1 ABLATION STUDY

For the SIREN's architecture, we evaluated the accuracy under various configurations including Spatial Self-Attention (SSA) only, with the Spiking State Space Model (S-SSM) module, and with Causal Temporal Self-Attention (CTSA) on the SL-Animals-DVS dataset, as summarized in Table 1. All results are averaged over 5 random seeds.

As shown in Table 1, with only spatial self-attention (SSA), average accuracy is 88.44%. Adding the S-SSM module increases accuracy to 91.01%. Further incorporating the CTSA module boosts accuracy to 92.89%.

Table 1: Ablations of SIREN.

| Conv SNN | SSA | S-SSM | CTSA | Acc (%) |
|:---:|:---:|:---:|:---:|:---|
| ✓ | | | | $77.33_{\pm 0.69}$ |
| ✓ | ✓ | | | $88.44_{\pm 0.10}$ |
| ✓ | | ✓ | | $87.38_{\pm 0.62}$ |
| ✓ | | | ✓ | $88.27_{\pm 0.17}$ |
| ✓ | ✓ | ✓ | | $91.01_{\pm 0.07}$ |
| ✓ | ✓ | | ✓ | $84.71_{\pm 1.62}$ |
| ✓ | | ✓ | ✓ | $89.51_{\pm 0.28}$ |
| ✓ | ✓ | ✓ | ✓ | $\mathbf{92.89}_{\pm 0.14}$ |

For exit techniques, to determine the optimal threshold, we use the Pareto frontier based on Loss vs. Steps and Accuracy vs. Steps to identify the theoretical optimal threshold. Building on this, we conduct experiments with three different scoring methods, as detailed in Appendix A.3. They are entropy, maximum confidence, and the difference between the maximum and second-highest confidence. The theoretical optimal threshold is identified as shown in Fig. 4. After comparing the knee points of the Pareto curves, we apply the Area Under the Curve (AUC) method to assess the most suitable scoring metric, ultimately finding that the entropy score yields the best performance among the three metrics.

## 5.2 ACCURACY UNDER FIXED TIME STEPS

We first evaluate classification accuracy under fixed inference steps to establish a baseline for comparison with prior work. Following common practice, we evaluate accuracy at multiple fixed steps (8, 10, and 16 steps) for direct comparison with approaches using different step configurations.

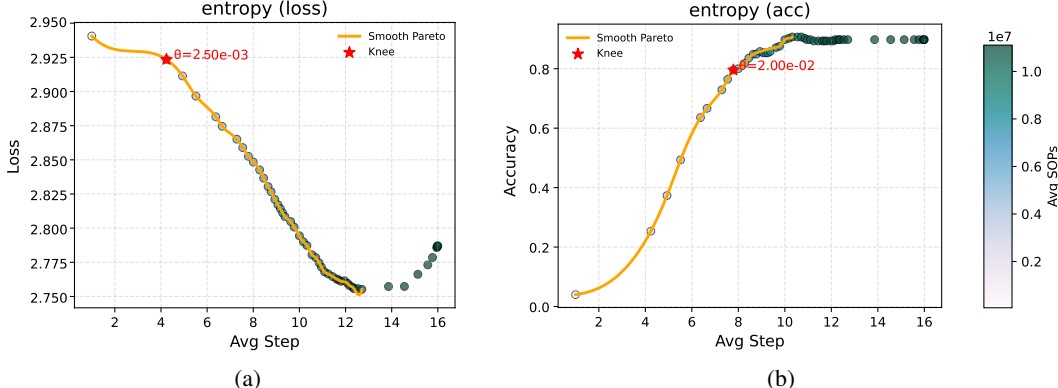

(a)                                                      (b)

Figure 4: The Pareto Front on SL-Animals-DVS, evaluated using the following metrics: (a) Loss vs. Steps. (b) Accuracy vs. Steps. $x$-axis is average exit step (lower is faster), $y$-axis is accuracy or loss (higher/lower is better). Colors encode synaptic operations (SOPs). Knee points highlight the optimal threshold $\theta$.

As shown in Table 2, SIREN consistently achieves state-of-the-art or comparable performance. On SL-Animals-DVS, it outperforms the best prior work by nearly 2% with fewer parameters, achieving over 90% accuracy in just 10 steps, while several other methods fail to reach this even at 16 steps. Similar advantages are observed on DVS128-Gesture and THU-EACT-50 subset, with the latter even reaching 100% accuracy. These results highlight our framework's top performance and efficiency with reduced time steps, making it ideal for deployment.

Table 2: Comparative results on neuromorphic datasets. *: self-implementation results with open source code. †: data augmentation. N/A: not reported or not applicable. Methods explicitly marked as (ANN) denote artificial neural networks, while unmarked methods correspond to spiking neural networks, bold represents the highest under the corresponding conditions.

| Dataset | Method | Architecture | Param.(M) | Steps | Acc.(%) |
|---|---|---|---|---|---|
| SL-Animals-DVS | TORE(Baldwin et al., 2023) | GoogLeNet(ANN) | 8.46 | 9 | 85.10 |
| | EvT(Sabater et al., 2022) | Transformer(ANN) | 0.47 | N/A | 88.12 |
| | SCTFA-SNN(Cai et al., 2024) | CNN | 3.13 | N/A | 90.04 |
| | TMC(Yan et al., 2025) | VGG-SNN | 9.40 | 16 | 70.05 |
| | EventRPG(Sun et al., 2024) | SEW ResNet-18 | 11.2 | 16 | 91.59 |
| | SpikeSlicer(Cao et al., 2024a) | ResNet-34 | 15.18 | N/A | 89.93 |
| | SGLFormer(Zhang et al., 2024) | Spiking Transformer | 4.17 | 16 | 88.44[*] |
| | Spike-SLR(Lin et al., 2024) | Spiking Transformer | 0.70 | 10 | 89.47 |
| | **Ours** | Spiking Transformer | 4.11 | 10 | 90.67 |
| | | Spiking Transformer | 4.11 | 16 | **93.33** |
| DVS128-Gesture | TORE(Baldwin et al., 2023) | GoogLeNet(ANN) | 8.46 | 9 | 96.20 |
| | EvT(Sabater et al., 2022) | Transformer(ANN) | 0.48 | N/A | 96.20 |
| | TMC(Yan et al., 2025) | VGG-SNN | 18.49 | 10 | 96.87[*] |
| | AGMM(Liang et al., 2025) | VGG-SNN | 9.22 | 16 | 97.92 |
| | EventRPG(Sun et al., 2024) | SEW ResNet-18 | 11.20 | 16 | 96.53 |
| | SpikeSlicer(Cao et al., 2024a) | ResNet-34 | 15.18 | N/A | 96.18 |
| | QKFormer(Chenlin Zhou, 2025) | Spiking Transformer | 1.50 | 16 | **98.60** |
| | SGLFormer(Zhang et al., 2024) | Spiking Transformer | 4.17 | 16 | 98.30[*] |
| | **Ours** | Spiking Transformer | 4.11 | 8 | 97.57 |
| | | Spiking Transformer | 4.11 | 16 | 97.92 |
| THU-EACT-50 | SGLFormer(Zhang et al., 2024) | Spiking Transformer | 4.17 | 16 | 99.75[*†] |
| | **Ours** | Spiking Transformer | 4.11 | 8 | 99.50[†] |
| | | Spiking Transformer | 4.11 | 16 | **100.00**[†] |

## 5.3 LATENCY EVALUATION

While the previous results demonstrate strong accuracy under fixed inference steps, practical deployment requires balancing performance with latency. In this subsection, we evaluate the trade-off between accuracy and inference steps.

By adjusting the threshold $\theta$, we enable the model to exit at different inference time steps. At every average time step from 1 to 16, we computed the model's inference accuracy, as shown in Figure 5(a), and the distribution of exit time steps, as depicted in Fig. 5(b), Fig. 5(c), and Fig. 5(d).

We mark the inflection point of the accuracy curve with a vertical dashed line in Fig. 5(a). On the DVS128-Gesture dataset, an average of 6.64 steps already achieves the accuracy of the full 16-step inference. For SL-Animals-DVS and THU-EACT-50, comparable accuracy is achieved at about 9.4 steps, requiring only half the inference steps of the 16-step baseline.

As shown in Fig. 5(b), Fig. 5(c), and Fig. 5(d), the exit time step distribution is compact under the same threshold $\theta$, with most samples completing prediction at similar steps. This demonstrates that the early-exit mechanism ensures stable, consistent behavior, minimizing latency fluctuations across samples.

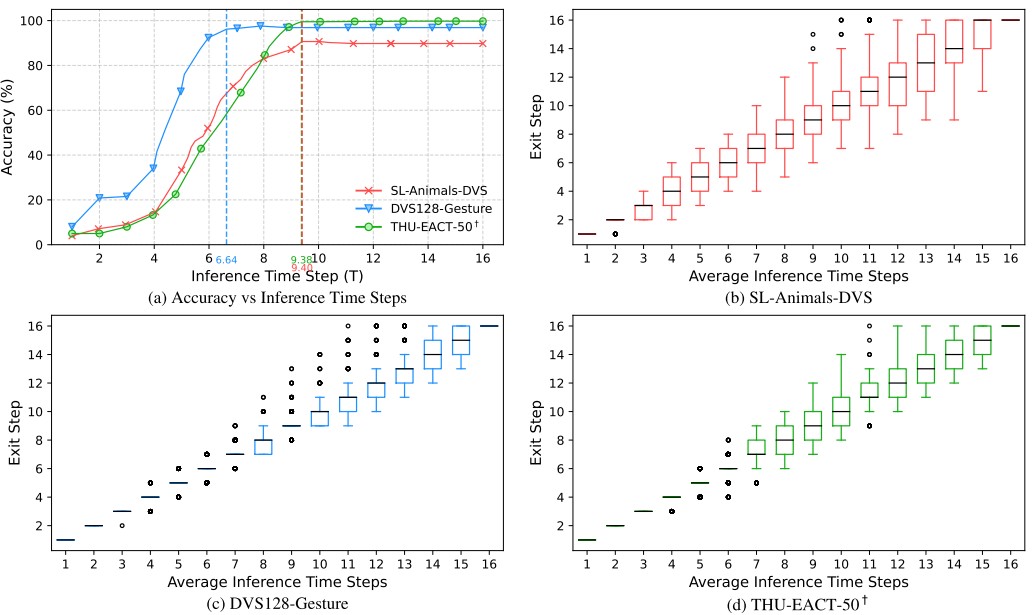

Figure 5: Trade-off between inference accuracy and time steps. (a) Accuracy vs. time steps by varying the threshold in Section 4.4, with dashed lines marking inflection points. Exit step distributions on (b) SL-Animals-DVS, (c) DVS128-Gesture, (d)THU-EACT-50 subset.

To further analyze the sensitivity of the accuracy–latency trade-off to hyperparameters and architecture, we vary the patience $\kappa$, EMA coefficient $\alpha$, network configurations, and confidence threshold $\theta$, and report the resulting accuracy–versus–average-exit-step curves in Fig. 6. The latency analysis provides evidence of efficient inference with reduced time steps. In the following subsection, we extend this evaluation to energy consumption, offering a more comprehensive view of the model's efficiency.

## 5.4 PRELIMINARY ENERGY EVALUATION

We aim to address challenges in deploying models on neuromorphic hardware. To evaluate energy efficiency, we measure FLOPs for ANNs and SOPs for SNNs, providing an estimate of computational costs. Using the same energy consumption calculation method in (Lin et al., 2024), we consider that every floating-point operation (FLOP) consumes 12.5 pJ, while each synaptic operation (SOP) requires 77 fJ. Energy is estimated from FLOPs/SOPs using standard per-op costs, and

hardware-in-the-loop neuromorphic measurements are left for future work. We therefore interpret the energy numbers as comparable proxies rather than absolute consumption.

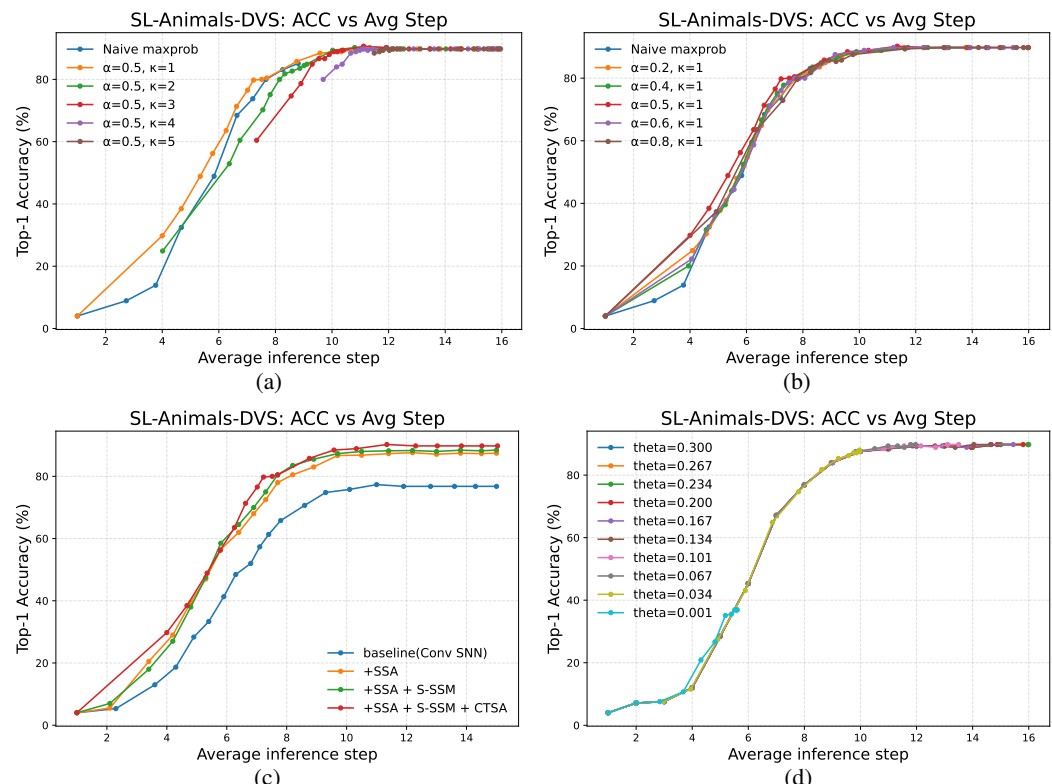

Figure 6: Sensitivity of Early-Exit behavior to hyperparameters and architecture on SL-Animals-DVS. (a) Effect of the patience parameter $\kappa$ in the early-exit rule. (b) Effect of the EMA coefficient $\alpha$ used for confidence smoothing. (c) Effect of model architecture (Conv SNN, +SSA, +SSA+S-SSM, and +SSA+S-SSM+CTSA). (d) Effect of the confidence threshold $\theta$ that controls the accuracy–latency trade-off.

Table 3: Computational Complexity and Energy Efficiency Comparison

| Method | Architecture | Param.(M) | FLOPs(G) | SOPs(G) | Energy(mJ) |
|---|---|---|---|---|---|
| TORE(Baldwin et al., 2023) | GoogLeNet(ANN) | 8.46 | 2.88 | N/A | 36.00 |
| TORE(Baldwin et al., 2023) | ResNet18(ANN) | 11.69 | 3.66 | N/A | 45.75 |
| EvT(Sabater et al., 2022) | Transformer(ANN) | 0.47 | 0.35 | N/A | 4.38 |
| Spike-SLR(Lin et al., 2024) | Spiking Transformer | 0.7 | N/A | 0.44 | N/A |
| AGMM(Liang et al., 2025) | N/A | 2.46 | 0.056 | 0.05 | **0.70** |
| **Ours** | Spiking Transformer | 4.11 | 0.102 | **0.02** | 1.28 |

Table 3 compares the computational complexity and energy efficiency of various methods. Our proposed SIREN shows exceptionally low synaptic operations, requiring only 0.02G SOPs and consuming just 1.28 mJ of energy during a full 16-step inference. This demonstrates its high sparsity and energy efficiency. Additionally, SIREN is expected to consume even less energy with fewer inference steps, which is encouraging for future low-power implementations.

## 6 CONCLUSION

This paper proposes an incremental inference framework for SNNs, serving as a rigorous and SNN-oriented framework for online inference. Our design incorporates the Spiking State-Space Model, Causal Time Self-Attention mechanism, and an early-exit mechanism based on the entropy of confidence. We achieve state-of-the-art accuracy on the DVS datasets with remarkably low synaptic operations. SIREN is a step towards hardware-aware, low-latency event-based recognition, with chip-level deployment left to future work.

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

# A  APPENDIX

In the main text, we outline the implementation of the incremental inference framework for spiking neural networks. A more detailed description of our method and experimental setup can be found in the appendix.

## A.1  MODEL DETAILS

### A.1.1  ADDITIONAL DETAILS FOR THE SPIKING STATE SPACE MODEL

Starting from the continuous-time diagonal form $\frac{d}{dt}\tilde{\boldsymbol{x}}(t) = \boldsymbol{\Lambda}\tilde{\boldsymbol{x}}(t) + \tilde{\boldsymbol{u}}(t)$, the ZOH recurrence in the main text Eq. 7 follows from the exact solution

$$\tilde{\boldsymbol{x}}_{k+1} = e^{\Delta\boldsymbol{\Lambda}}\tilde{\boldsymbol{x}}_k + \Big( \int_0^\Delta e^{\tau\boldsymbol{\Lambda}}\, d\tau \Big) \tilde{\boldsymbol{u}}_k,$$

$$\int_0^\Delta e^{\tau\boldsymbol{\Lambda}}\, d\tau = \boldsymbol{\Lambda}^{-1}\big(e^{\Delta\boldsymbol{\Lambda}} - \boldsymbol{I}\big) \tag{17}$$

because $\boldsymbol{\Lambda}$ is diagonal, Eq. 7 decouples across modes, yielding the modal update in Eq. 9 and the well-conditioned limit $\bar{s}_i \to \Delta$ when $\lambda_i \to 0$.

When $\lambda_i = -\alpha_i + i\omega_i$ with $\alpha_i > 0$, one may avoid complex arithmetic by using the real $2 \times 2$ representation

$$\boldsymbol{x}_{i,k+1} = e^{-\alpha_i\Delta} \begin{bmatrix} \cos(\omega_i\Delta) & -\sin(\omega_i\Delta) \\ \sin(\omega_i\Delta) & \cos(\omega_i\Delta) \end{bmatrix} \boldsymbol{x}_{i,k} \; + \; \boldsymbol{s}_i\,\tilde{\boldsymbol{u}}_{i,k} \tag{18}$$

where $\boldsymbol{x}_{i,k} = [x_{i,\mathrm{re},k},\, x_{i,\mathrm{im},k}]^\top$ and $\boldsymbol{s}_i = [\Re(\bar{s}_i),\, \Im(\bar{s}_i)]^\top$. This form is algebraically equivalent to the complex update.

Causality is immediate since $\tilde{\boldsymbol{x}}_{k+1}$ depends only on past inputs and $\tilde{\boldsymbol{x}}_k$. For BIBO stability under ZOH, the eigenvalues of $\bar{\boldsymbol{N}}$ equal $e^{\Delta\lambda_i}$, thus $|e^{\Delta\lambda_i}| = e^{\Delta\,\Re(\lambda_i)} < 1$ whenever $\Re(\lambda_i) < 0$, implying $\rho(\bar{\boldsymbol{N}}) < 1$. This refines the stability condition stated in the main text.

We use per-mode steps $\Delta_i$ (with a global rescale), so that $\bar{n}_i = e^{\Delta_i\lambda_i}$ and $\bar{s}_i = (e^{\Delta_i\lambda_i} - 1)/\lambda_i$. For $|\Delta_i\lambda_i| \ll 1$, using $\mathrm{expm1}$ for $e^z - 1$ and the limit $\bar{s}_i \approx \Delta_i$ improves numerical robustness.

For event data, the recurrence in Eq. 7 is applied incrementally with state carry-over and $\tilde{\boldsymbol{x}}_0 = \boldsymbol{0}$ at sequence start, giving $\mathcal{O}(H)$ compute and constant memory per step. If inputs are modeled as Dirac impulses, the exact solution yields a Dirac discretization that is preferable when events are localized, otherwise ZOH is exact for piecewise-constant frames.

Even if $\tilde{\boldsymbol{x}}_k$ is complex, real outputs follow by design via conjugate-symmetric modes, real-valued spiking heads (Cartesian or polar), or taking $\Re(\cdot)$ before the spiking nonlinearity.

### A.1.2  DETAILS FOR STATEFUL CAUSAL SPATIAL-TEMPORAL ATTENTION

Let $\boldsymbol{X}_t \in \mathbb{R}^C$ be intermediate features. Spike-generating maps $\mathcal{S}_Q, \mathcal{S}_K, \mathcal{S}_V$ produce

$$\boldsymbol{Q}_t = \mathcal{S}_Q(\boldsymbol{X}_t), \qquad \boldsymbol{K}_{t'} = \mathcal{S}_K(\boldsymbol{X}_{t'}), \qquad \boldsymbol{V}_{t'} = \mathcal{S}_V(\boldsymbol{X}_{t'}) \tag{19}$$

Channels are split into $h$ heads of width $d_h$ ($C = h\,d_h$). With a causal mask

$$\boldsymbol{M}_{t,t'} = \begin{cases} 1, & t' \le t \\ 0, & t' > t, \end{cases} \tag{20}$$

and a per-head scale

$$\sigma = \frac{\zeta_h}{\sqrt{d_h}}, \qquad \zeta_h > 0 \tag{21}$$

the temporal scores and aggregation are

$$\beta_{t,t'} = \sigma\,\boldsymbol{Q}_t\,\boldsymbol{K}_{t'}^\top\,\boldsymbol{M}_{t,t'}, \qquad \mathrm{CTSA}_t = \Theta\Big( \sum_{t' \le t} \beta_{t,t'}\,\boldsymbol{V}_{t'} \Big) \tag{22}$$

where $\Theta(\cdot)$ is a spiking activation (e.g., LIF).

To make magnitudes insensitive to the number of eligible past steps, define

$$Z_t = \sum_{t' \leq t} \boldsymbol{M}_{t,t'}, \quad w_{t,t'} = \frac{\beta_{t,t'}}{\max(1, Z_t)}, \quad \text{CTSA}_t = \Theta\Big(\sum_{t' \leq t} w_{t,t'}\, \boldsymbol{V}_{t'}\Big) \tag{23}$$

Let $s_t = \|\boldsymbol{Q}_t\|_0$ and $s_{t'} = \|\boldsymbol{K}_{t'}\|_0$ denote the number of active channels of the head at steps $t$ and $t'$, respectively, so $0 \leq s_t, s_{t'} \leq d_h$. If either (A) unit-norm features per head hold, $\|\boldsymbol{Q}_t\|_2 = \|\boldsymbol{K}_{t'}\|_2 = 1$, or (B) spikes are bounded with supports $\|\boldsymbol{Q}_t\|_2 \leq \sqrt{s_t}$, $\|\boldsymbol{K}_{t'}\|_2 \leq \sqrt{s_{t'}}$ and $s_t, s_{t'} \leq d_h$, then by Cauchy–Schwarz

$$|\beta_{t,t'}| \leq \begin{cases} \sigma, & \text{(A)} \\ \sigma\sqrt{s_t s_{t'}}, & \text{(B)} \end{cases} \tag{24}$$

Thus scores are uniformly bounded under (A), or scale only with the number of active channels under (B). The factor $\sigma = \zeta_h/\sqrt{d_h}$ prevents growth with $d_h$.

With equation 23,

$$\Big\|\sum_{t' \leq t} w_{t,t'}\, \boldsymbol{V}_{t'}\Big\| \leq \frac{1}{Z_t}\sum_{t' \leq t} |\beta_{t,t'}|\, \|\boldsymbol{V}_{t'}\| \leq \begin{cases} \sigma\, \overline{v}_t, & \text{(A)} \\ \sigma\, \overline{v}_t\, \overline{s}_t, & \text{(B)} \end{cases} \tag{25}$$

where $\overline{v}_t = \frac{1}{Z_t}\sum_{t' \leq t}\|\boldsymbol{V}_{t'}\|$ and $\overline{s}_t = \frac{1}{Z_t}\sum_{t' \leq t}\sqrt{s_t s_{t'}}$. Hence the aggregate magnitude does not grow with history length $Z_t$.

Consider the linear operator $\boldsymbol{W}_t : \{\boldsymbol{V}_{t'}\}_{t' \leq t} \mapsto \sum_{t' \leq t} w_{t,t'}\boldsymbol{V}_{t'}$. Under case (A) with equation 23, the weight bound gives $\|\boldsymbol{W}_t\|_2 \leq \sigma$. If $\Theta$ is 1-Lipschitz in its linear region (standard surrogate), then

$$\|\text{CTSA}_t\| \leq \sigma\, \big\|\{\boldsymbol{V}_{t'}\}_{t' \leq t}\big\|, \qquad \|\nabla\text{CTSA}_t\| \leq \sigma \tag{26}$$

which controls forward and backward norms without softmax.

Let $r_Q = \mathbb{E}[\|\boldsymbol{Q}_t\|_0/d_h]$ and $r_K = \mathbb{E}[\|\boldsymbol{K}_{t'}\|_0/d_h]$ be per-head firing rates. Under a simple independence model

$$\mathbb{E}\Big[\, \big|\{\, t' \leq t : \beta_{t,t'} \neq 0\,\}\big|\,\Big] \approx r_Q\, r_K\, Z_t \tag{27}$$

so the per-token temporal cost scales as $\mathcal{O}(r_Q r_K Z_t d_h)$ instead of $\mathcal{O}(Z_t d_h)$ for dense attention.

Use $\sigma = \zeta_h/\sqrt{d_h}$ with $\zeta_h$ initialized to 1 (e.g., via a positive parameterization). Either apply vectorwise $\ell_2$ normalization (case A) or rely on bounded spike supports (case B). Optionally clip $\beta_{t,t'}$ to $[-c, c]$ for extra robustness.

For reference, standard self-attention with softmax uses

$$\alpha_{t,t'}^{\text{soft}} = \frac{\exp(\gamma\, \boldsymbol{Q}_t \boldsymbol{K}_{t'}^\top\, \boldsymbol{M}_{t,t'})}{\sum_{s \leq t}\exp(\gamma\, \boldsymbol{Q}_t \boldsymbol{K}_s^\top\, \boldsymbol{M}_{t,s})}, \qquad \text{Attn}_t^{\text{soft}} = \sum_{t' \leq t}\alpha_{t,t'}^{\text{soft}}\, \boldsymbol{V}_{t'} \tag{28}$$

with a temperature $\gamma > 0$. Here the weights $\alpha_{t,t'}^{\text{soft}}$ form a probability distribution over past steps $t' \leq t$, which enforces strong competition: a few positions can dominate, and every new token changes the normalization over all previous ones.

By contrast, CTSA replaces the softmax normalization with bounded scores $\beta_{t,t'}$ and length-normalized weights $w_{t,t'}$ in equation 23. Intuitively, preserves the sign and relative magnitude of the dot products $\boldsymbol{Q}_t \boldsymbol{K}_{t'}^\top$ and avoids global renormalization as $t$ grows. Together with the bounds above, this yields a streaming-friendly temporal operator whose forward and backward norms are controlled without softmax, and whose cost scales as $\mathcal{O}(r_Q r_K Z_t d_h)$ instead of $\mathcal{O}(Z_t d_h)$ for dense attention.

In terms of when each is preferable, CTSA is particularly suitable in our setting of incremental, low-latency event processing: it naturally supports causal, stepwise updates, sparse spikes, and bounded temporal cost without recomputing a softmax over a growing window. Standard softmax attention may still be advantageous in large, offline transformers where sharp, highly selective attention maps

are desired and energy or latency constraints are less critical, as its probability simplex can yield more peaked focus.

A limitation of CTSA is that it does not produce a probability distribution over time and is more sensitive to the overall scale of $\boldsymbol{Q}_t$ and $\boldsymbol{K}_{t'}$: there is no implicit global normalization as in softmax. If feature scales are poorly controlled, CTSA can be less robust and may underperform softmax in purely accuracy-driven, high-capacity regimes. In SIREN we mitigate this with the per-head scaling $\sigma$, normalization of features (case (A) or (B) above), and bounded effective history length $Z_t$, which empirically suffices on the DVS benchmarks considered in this work.

### A.1.3 EXIT MECHANISM

At time step $t$, the readout produces class evidence $\boldsymbol{z}_t \in \mathbb{R}^C$. We form temperature-scaled probabilities:

$$\boldsymbol{p}_t = \mathrm{softmax}\big(\boldsymbol{z}_t/T\big), \qquad p_t(c) \;=\; \frac{\exp(z_t(c)/T)}{\sum_{c'} \exp(z_t(c')/T)} \tag{29}$$

where, $c \in 1, ..., C$ denotes the class index, and $\boldsymbol{p}_t$ the class–probability vector with component $p_t(c)$. To suppress stepwise jitter, we apply exponential moving average (EMA) in probability space:

$$\widehat{\boldsymbol{p}}_t = \alpha\,\widehat{\boldsymbol{p}}_{t-1} + (1 - \alpha)\,\boldsymbol{p}_t, \qquad \widehat{\boldsymbol{p}}_0 = \boldsymbol{p}_0 \tag{30}$$

where $\alpha$ denotes the smoothness coefficient. The predicted class is $\widehat{y}_t = \arg\max_c \widehat{p}_t(c)$. We define three confidence scores $s_t$:

$$s_t^{\mathrm{max}} = \max_c \widehat{p}_t(c) \tag{31}$$

$$s_t^{\mathrm{mar}} = \widehat{p}_t(c_1) - \widehat{p}_t(c_2) \tag{32}$$

$$s_t^{\mathrm{ent}} = 1 - \frac{H(\widehat{\boldsymbol{p}}_t)}{\log C}, \qquad H(\widehat{\boldsymbol{p}}_t) = -\sum_{c=1}^{C} \widehat{p}_t(c)\,\log \widehat{p}_t(c) \tag{33}$$

where $c_1, c_2$ are the top-1 and top-2 classes of $\widehat{\boldsymbol{p}}_t$.

We adopt a stability–patience rule to avoid premature exits. Let $\theta$ be a criterion-specific threshold and $\kappa$ the patience. We stop at

$$t^{\star} = \min\Big\{t: \; s_t \geq \theta \text{ for } \kappa \text{ consecutive steps and } \widehat{y}_{t-k} = \widehat{y}_t, \; \forall\, k \in \{1, \ldots, \kappa - 1\}\Big\} \tag{34}$$

**Model–efficiency selection via AUC.** To compare criteria, we sweep $\theta$ to obtain accuracy–cost pairs $\{(x_i, y_i)\}$, where $x_i$ is average time steps and $y_i$ is accuracy or loss. We keep the Pareto-efficient set sorted by $x$ and compute a scale-free area under the curve (AUC) on the normalized axes:

$$\tilde{x}_i = \frac{x_i - \min_j x_j}{\max_j x_j - \min_j x_j}, \qquad \tilde{y}_i = \frac{y_i - \min_j y_j}{\max_j y_j - \min_j y_j}, \tag{35}$$

$$\mathrm{AUC} \approx \sum_i \frac{1}{2}\,(\tilde{y}_i + \tilde{y}_{i+1})\,(\tilde{x}_{i+1} - \tilde{x}_i) \tag{36}$$

A larger AUC indicates a better accuracy–efficiency trade-off across exit settings, and we select the criterion with the highest AUC.

## A.2 EXPERIMENT SETTINGS

### A.2.1 DATASETS AND PREPROCESSING

**SL-Animals-DVS:** The SL-Animals-DVS dataset (Vasudevan et al., 2022) focuses on sign language recognition, featuring 19 classes of animal-related signs. It comprises approximately 1,100 samples captured using a Dynamic Vision Sensor (DVS) with a resolution of 128×128.

**DVS128-Gesture:** DVS128-Gesture (Amir et al., 2017) is designed for hand gesture recognition, containing recordings of 10 or 11 different hand gestures with a resolution of 128×128. The dataset includes 1,342 instances grouped into 122 trials, collected from 29 subjects under three different lighting conditions.

**THU-EACT-50:** THU-EACT-50 (Gao et al., 2023) is a large-scale, real-world event-based action recognition dataset comprising 50 action categories performed by 105 subjects. It includes 10,500 video recordings captured using the CeleX-V event camera at a resolution of 1280×800 pixels. For computational efficiency, we selected 20 classes from the original THU-EACT-50 dataset due to resource constraints and downsampled the event frames to a resolution of 128×128 pixels. We also report results on the full set where feasible or justify the subset due to compute limits.

Table 4: Comparison of event-based datasets used in our experiments

| Dataset | Classes | Recordings | Resolution | Avg. Duration / Conditions |
|---|---|---|---|---|
| SL-Animals-DVS | 19 | ∼1,100 | 128×128 | Varying illumination, isolated sign gestures |
| DVS128-Gesture | 11 | 1,342 | 128×128 | Three lighting conditions; durations 1–6s |
| THU-EACT-50 | 50 | 10,500 | 1280×800 | Real-world actions; indoors; various subjects |

**Data Preprocessing:** We aggregated the events in the dataset into event frames at equal time intervals, with each sample divided into 16 frames. For computational efficiency, and due to resource constraints, we selected 20 classes from the original THU-EACT-50 dataset. Additionally, we downsampled the event frames to a resolution of 128×128 pixels to further optimize processing time and memory usage.

### A.2.2 TRAINING SETTINGS

We summarize the training configurations used in all experiments as shown in Table 5.

We use the AdamW optimizer with a weight decay of 0.06 and an initial learning rate of 0.005, enabling automatic mixed precision (AMP) and synchronized batch normalization. A cosine decay learning rate schedule is adopted with 10 warmup epochs and 10 cooldown epochs, and the total number of training epochs is 150. Label smoothing ($\epsilon = 0.1$) is applied, Mixup ($\gamma_1 = 0.5$) is enabled with probability 0.5, while CutMix is disabled. Training is conducted with a batch size of 8 and 8 workers for data loading, TensorBoard is used for logging, and evaluation is performed using the checkpoint with the best validation accuracy.

## A.3 EXPERIMENT

### A.3.1 PARETO FRONTIER AND AUC ANALYSIS OF MODEL TRADE-OFFS

We evaluated the model using two key metrics: Accuracy vs. Step and Loss vs. Step. To analyze the trade-offs, we plotted the Pareto frontier based on three scoring criteria: (1) cross-entropy, (2) maximum confidence, and (3) the difference between the maximum and second-highest confidence scores. Each point on the Pareto frontier is color-coded to represent the corresponding synaptic operation (SOP) value, as illustrated in Fig. 7.

The AUC method was used to compare the three scoring metrics, and entropy was found to be the optimal score, as shown in Table 7. This table compares the early-exit decision methods, with the best values highlighted in bold.

Table 5: Training settings.

| Parameter | Value |
|---|---|
| Batch size | 8 |
| Epochs | 150 |
| Optimizer | AdamW |
| Learning rate | 0.005 |
| Weight decay | 0.06 |
| Scheduler | Cosine decay |
| Warmup epochs | 10 |
| Cooldown epochs | 10 |
| Label smoothing | 0.1 |
| Mixup $\gamma_1$ | 0.5 |
| CutMix $\gamma_2$ | 0.0 |
| AMP training | Enabled |

Table 6: Hyperparameters used in the Early-Exit Gate and Inference Process.

| Hyperparameter | Value |
|---|---|
| Threshold ($\tau$) | 0.8 |
| Patience ($\kappa$) | 1 |
| EMA Coefficient ($\alpha$) | 0.8 |
| Temperature ($T$) | 1.0 |
| Exit Score (score) | entropy |

### A.3.2 PERFORMANCE COMPARISON ON ADDITIONAL NEUROMORPHIC DATASETS

We further evaluate our model on two more challenging neuromorphic benchmarks, CIFAR10-DVS and N-Caltech101. As shown in Table 8, under the same 16-step setting, our ChronoSpik-Former achieves comparable performance to recent spiking Transformers on CIFAR10-DVS (79.6% vs. 80.9% for Spikformer and 82.9% for QKFormer), while attaining the best accuracy on N-Caltech101, reaching 84.0% and surpassing QKFormer (83.6%) and SpikingResformer (81.3%). These results indicate that our architecture scales well to more complex event-based recognition tasks and remains competitive with state-of-the-art spiking Transformers. For all compared methods in Table 8, we directly report the results from Lee et al. (2025).

### A.3.3 ACCURACY AND SYNAPTIC OPERATIONS (SOPs) ACROSS INFERENCE STEPS

Table 9, Table 10 and Table 11 present the relationship between accuracy and synaptic operations (SOPs) across three datasets: SL-Animals-DVS, DVS128-Gesture, and THU-EACT-50. For each dataset, we evaluate the trade-off by varying the threshold and measuring the accuracy and SOPs at different inference time steps. These tables illustrate how adjusting the threshold impacts both the model's recognition performance (accuracy) and computational efficiency (SOPs) under different conditions.

### A.4 LIMITATIONS

Our primary goal in this work is to tackle challenging temporal modeling in event-based recognition, and SIREN is specifically designed to exploit rich, non-trivial dynamics over time. When a task does not require complex temporal reasoning and performance is instead dominated by spatial modeling, the advantages of SIREN become less pronounced. For example, on CIFAR10-DVS and N-Caltech101, although the data are recorded with real DVS sensors, they are constructed by mounting the camera on a tripod and sweeping a monitor displaying static images. This acquisition protocol induces relatively simple and stereotyped temporal patterns and places a stronger emphasis on spatial representation. As a result, these benchmarks do not fully showcase the strengths of

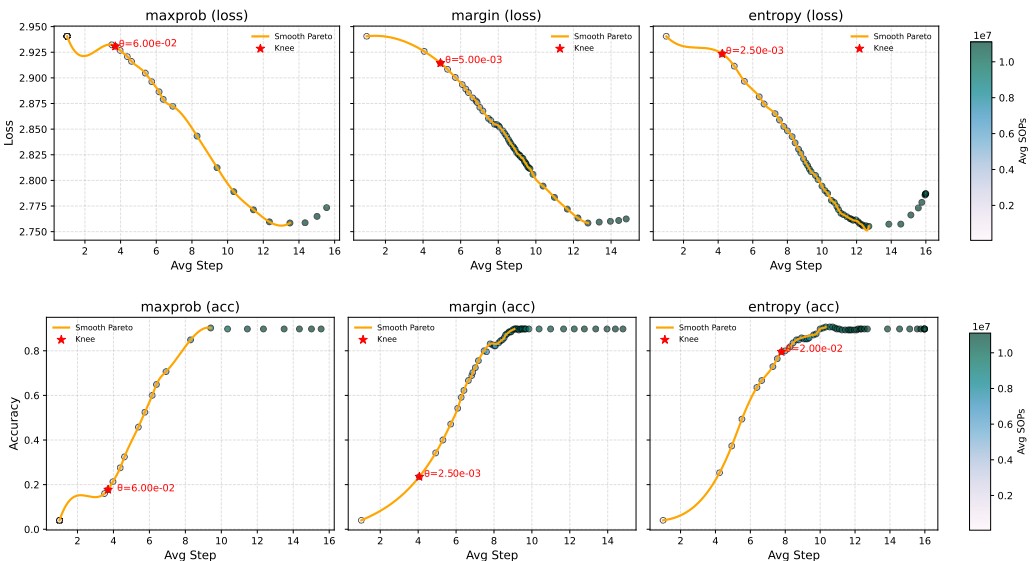

Figure 7: Pareto Frontier Analysis for Accuracy vs. Step and Loss vs. Step Metrics: the Pareto frontier plotted using three scoring criteria: (1) cross-entropy, (2) maximum confidence, and (3) the difference between the maximum and second-highest confidence scores. Each point on the frontier is color-coded to represent the corresponding synaptic operations (SOPs) value, showcasing the trade-offs between accuracy, loss, and computational efficiency.

Table 7: Comparison of early-exit decision methods. The best values are highlighted in bold.

| Method | AUC ↑ | Knee ($\theta$) | Acc ↑ | Step ↓ | Loss ↓ |
|---|---|---|---|---|---|
| entropy | **0.6995** | 0.020 | **0.7956** | 7.79 | **2.8526** |
| maxprob | 0.6934 | 0.060 | 0.1778 | 3.70 | 2.9308 |
| margin | 0.6816 | 0.003 | 0.2356 | 4.06 | 2.9258 |

SIREN's temporal modeling, and our method provides only modest gains compared to architectures with stronger spatial backbones. Our current implementation assumes that the raw event stream is first aggregated into a fixed number of frames (16 in all experiments), and incremental inference is performed over this frame sequence. This choice follows the dominant evaluation protocol for DVS benchmarks but does not yet realize truly event-driven processing at the level of individual timestamps. Moreover, our energy analysis is based on generic FLOPs/SOPs proxies rather than hardware-in-the-loop measurements on a specific neuromorphic platform. Extending SIREN to operate directly on raw event streams and validating it on concrete neuromorphic chips are important directions for future work.

## A.5 LARGE LANGUAGE MODELS USAGE

We used large language models (LLMs) to polish writing. Specifically, we use ChatGPT (OpenAI, GPT-5) to (i) polish grammar and phrasing of paragraphs drafted by the authors, and (ii) suggest alternative expressions for figure captions and section headings. The models did not generate novel technical ideas, algorithms, experimental designs, or results, all scientific content, claims, and analysis are created and verified by the authors.

Table 8: Comparative results on CIFAR10-DVS and N-Caltech101. Methods in bold denote our proposed models.

| Dataset | Method | Architecture | Steps | Acc.(%) |
|---|---|---|---|---|
| CIFAR10-DVS | QKFormer(Chenlin Zhou, 2025) | QKFormer-4-384 | 16 | **82.9** |
| | SpikingResformer(Shi et al., 2024) | SpikingResformer-Ti | 16 | 78.8 |
| | Spikformer(Zhou et al., 2023) | Spiking Transformer-2-256 | 16 | 80.9 |
| | **Ours** | ChronoSpikFormer-2-256 | 16 | 79.6 |
| N-Caltech101 | QKFormer(Chenlin Zhou, 2025) | QKFormer-4-384 | 16 | 83.6 |
| | SpikingResformer(Shi et al., 2024) | SpikingResformer-Ti | 16 | 81.3 |
| | Spikformer(Zhou et al., 2023) | Spikformer-4-384 | 16 | 75.1 |
| | **Ours** | ChronoSpikFormer-3-256 | 16 | **84.0** |

Table 9: Accuracy (%) and SOPs over 16 time steps for SL-Animals-DVS

| Time Step | ACC (%) | SOPs (M) |
|---|---|---|
| 1 | 4.00 | 0.05 |
| 2 | 7.11 | 0.25 |
| 3 | 8.89 | 0.87 |
| 4 | 14.67 | 1.60 |
| 5 | 33.33 | 2.48 |
| 6 | 47.56 | 3.44 |
| 7 | 61.33 | 5.68 |
| 8 | 70.67 | 8.95 |
| 9 | 77.33 | 9.14 |
| 10 | 83.11 | 11.64 |
| 11 | 87.11 | 13.72 |
| 12 | 90.67 | 16.54 |
| 13 | 90.67 | 16.97 |
| 14 | 89.78 | 17.94 |
| 15 | 89.78 | 18.03 |
| 16 | 89.78 | 18.11 |

Table 10: Accuracy (%) and SOPs over 16 time steps for DVS128-Gesture

| Time Step | ACC (%) | SOPs (M) |
|---|---|---|
| 1 | 8.00 | 0.06 |
| 2 | 20.83 | 0.28 |
| 3 | 21.53 | 0.70 |
| 4 | 34.03 | 1.33 |
| 5 | 41.67 | 2.18 |
| 6 | 51.39 | 3.27 |
| 7 | 57.99 | 4.64 |
| 8 | 68.40 | 6.07 |
| 9 | 76.04 | 7.30 |
| 10 | 87.15 | 7.79 |
| 11 | 92.36 | 8.08 |
| 12 | 96.18 | 8.88 |
| 13 | 96.53 | 9.49 |
| 14 | 96.88 | 10.10 |
| 15 | 96.88 | 10.60 |
| 16 | 96.88 | 10.73 |

Table 11: Accuracy (%) and SOPs over 16 time steps for THU-EACT-50

| Time Step | ACC (%) | SOPs (M) |
|---|---|---|
| 1 | 5.00 | 0.03 |
| 2 | 5.00 | 0.19 |
| 3 | 8.00 | 0.48 |
| 4 | 13.25 | 0.99 |
| 5 | 22.50 | 2.12 |
| 6 | 42.88 | 3.09 |
| 7 | 55.13 | 4.37 |
| 8 | 67.87 | 5.32 |
| 9 | 78.63 | 6.07 |
| 10 | 84.63 | 6.96 |
| 11 | 91.88 | 7.62 |
| 12 | 96.63 | 8.51 |
| 13 | 97.13 | 9.46 |
| 14 | 98.75 | 10.02 |
| 15 | 99.50 | 10.28 |
| 16 | 99.75 | 10.31 |

