# OpenReview forum: "Decide When Ready: Stepwise Incremental Inference with Early-Exit in Spiking Neural Networks"
_ICLR.cc/2026/Conference — Submitted to ICLR 2026_

### Official Review · Reviewer_KzPq · 2025-10-23

**Soundness:** 2
**Presentation:** 1
**Contribution:** 2
**Rating:** 2
**Confidence:** 5

**Summary:**

This paper proposes a spiking neural network (SNN) framework that integrates LIF and RF neurons with a spiking state-space module (S-SSM). Additionally, this paper proposes causal spatial-temporal self-attention (CSTSA) and uses an early-exit mechanism to improve the inference efficiency. The author conducted experiments on three datasets and compared the results with other methods.

**Strengths:**

The experimental results show that the proposed method has advantages over other methods.

**Weaknesses:**

1. **Unclear presentation**. Although the authors introduced the proposed S-SSM, C-STSA, and the early exit mechanism, Figure 2 indicates that their method also uses SpatialFormer and TemporalFormer, which are not described. Additionally, the author refers to the proposed method as SIREN. What, then, is ChronoSpikFormer in Figure 2? There is also no textual description of its overall architecture.

2. **High complexity**. The proposed S-SSM is significantly more complex than the commonly used LIF neuron, which makes the proposed method difficult to implement. Additionally, Equation (14) shows that C-STSA introduces real-valued multiplication operations that are incompatible with the spike-driven characteristics of SNNs. Therefore, the advantages of the proposed method are still only theoretical.

3. **Insufficient experiments**.

* The performance reported in this paper on the three datasets is nearly saturated. To demonstrate their method's effectiveness, the authors should conduct experiments on more challenging datasets, such as CIFAR10-DVS, N-Caltech101, and ImageNet.

* Furthermore, existing SNNs on DVS128-Gesture operate at a resolution of $48\times48$. However, this paper uses a resolution of $128\times128$, which makes for an unfair comparison.

* Existing methods have achieved better performance than the present work. For example, QKFormer [1] attained an accuracy of 98.6% on the DVS-Gesture dataset. Therefore, the authors should compare their method with the latest methods.

* To demonstrate its advantages, the author should compare their method with other early exit mechanisms.

[1] QKFormer: Hierarchical Spiking Transformer using Q-K Attention. In NeurIPS. 2024.

**Questions:**

Could the author explain why using entropy outperforms the maximum probability and marginal metrics by such a significant margin?

---

> ### Author Response · Authors · 2025-11-19
> **Response to Reviewer KzPq (part 1)**
>
> Thank you for taking the time to review. Please see our detailed response to the issues you have raised.
> ## Response to Weaknesses
> ***W1: Unclear model presentation: Figure 2 includes SpatialFormer and TemporalFormer, which are not explained; the text names the method SIREN, while Figure 2 uses the term ChronoSpikFormer; and there is no clear textual description of the overall architecture.***
>
> **ANSWER:** We thank the reviewer for pointing out this confusion, this is indeed due to our oversight in the current manuscript. In the **revised Figure 2 (Sec. 4.1)**, we have replaced the labels *SpatialFormer* and *TemporalFormer* with the terminology used in the main text: **SSA (Spatial Self-Attention)** and **CTSA (Causal Time Self-Attention)**. We clarify that **ChronoSpikFormer** denotes the *backbone spiking architecture* (Stem + Tokenizer + S-SSM + SSA + CTSA), **without** the early-exit mechanism. **SIREN** refers to the *full method*: ChronoSpikFormer **plus** the incremental inference data input and early-exit strategy.
>
> We have also added a concise textual description of the overall architecture in the caption of Figure 2. The description is as follows.
>
> The ChronoSpikFormer backbone consists of a stem and tokenizer implemented with convolutional operations, followed by the spatial self-attention (SSA) module, the causal time self-attention (CTSA) module, and the spiking state-space model (S-SSM) module consisting of SSM and RF spiking technique. During training, SIREN operates with fixed time steps. During inference, event frames $F_i$ arrive sequentially and incrementally update the S-SSM state $X_i$, producing class probabilities $P_i$. The entropy score of $P_i$ is monitored at each step, and once it exceeds a predefined threshold, SIREN performs early exit at time step $i$.
>
> ***W2: The reviewer questions whether the added complexity of S-SSM and C-STSA is practical for neuromorphic hardware, suggesting that the claimed advantages might not translate into real deployments.***
>
> **ANSWER:** We thank the reviewer for raising this concern. Implementation complexity is indeed important for neuromorphic models, but our contribution is algorithmic rather than a chip design. Both S-SSM and C-STSA are, however, built from operations already supported on existing neuromorphic hardware, so they are not purely theoretical.
>
> **S-SSM.**
> All continuous-time parameters are pre-discretized offline, so at inference S-SSM reduces to a per-step linear recurrence $x_{k+1} = \bar{N} x_k + \bar{S} u_k$ followed by a spiking nonlinearity. Compared with a standard (non-spiking) SSM, we simply replace the continuous activation (e.g., ReLU) with a threshold-and-reset mechanism and interpret $x_k$ as membrane state, which matches the native computation of Loihi cores (integer-weighted accumulations, leak, local state, spike emission)$^{[1]}$. Each S-SSM mode is a small real-valued recurrence (1D or a $2\times 2$ block for complex modes) plus a local state register, i.e., a fixed-coefficient digital filter attached to a spiking neuron. Diagonal / block-diagonal SSMs with this structure have already been implemented efficiently on Loihi 2 for streaming sequence processing (e.g., S4D)$^{[2]}$, and sparse linear RNNs show large latency and energy gains on Loihi 2 compared to dense models on edge GPUs$^{[3]}$. Together with work on compiling dynamical systems into spiking hardware$^{[4]}$, this suggests that S-SSM–like modules are implementable and can be efficient on Loihi-like accelerators.
>
> **C-STSA.**
> C-STSA computes dot products between spike-derived queries and keys and accumulates weighted values using fixed-precision MACs and thresholds, without softmax. Such MACs and local state updates are standard primitives on digital neuromorphic chips (e.g., Loihi with programmable integer-weighted synapses and on-chip learning)$^{[1]}$. Thus C-STSA can be viewed as a sparsified, spike-gated attention mechanism built from existing low-level operations, rather than requiring qualitatively new hardware.
>
> [1] Davies, Mike, et al. "Loihi: A neuromorphic manycore processor with on-chip learning." Ieee Micro, 2025.
> [2] Meyer, Svea Marie, et al. "A diagonal structured state space model on loihi 2 for efficient streaming sequence processing." 2025 NICE.
> [3] Pierro, Alessandro, et al. "Accelerating Linear Recurrent Neural Networks for the Edge with Unstructured Sparsity." arXiv preprint arXiv:2502.01330 ,2025.
> [4] Voelker, Aaron Russell. Dynamical systems in spiking neuromorphic hardware. Diss. University of Waterloo, 2019.
> [5] Carney, Rebecca, et al. "Neuromorphic Kalman filter implementation in IBM’s TrueNorth." Journal of Physics: Conference Series., 2017.
> [6] S. K. Esser, P. A. Merolla, J. V. Arthur, et al. "Convolutional networks for fast, energy-efficient neuromorphic computing." (PNAS), 2016.

---

> > ### Author Response · Authors · 2025-11-26
> > **Correction on Loihi Reference**
> >
> > In our rebuttal, we mistakenly cited Loihi as “Davies et al., 2025”. The correct reference is
> >
> > [1] M. Davies et al., "Loihi: A Neuromorphic Manycore Processor with On-Chip Learning," in IEEE Micro, vol. 38, no. 1, pp. 82-99, 2018.
> >
> > We apologize for the oversight.

---

> ### Author Response · Authors · 2025-11-19
> **Response to Reviewer KzPq (part 2)**
>
> This is a continued response to the weakness 3.
> ***W3: Insufficient experiments.***
> ***The performance reported in this paper on the three datasets is nearly saturated. To demonstrate their method's effectiveness, the authors should conduct experiments on more challenging datasets, such as CIFAR10-DVS, N-Caltech101, and ImageNet.***
>
> **ANSWER:** We thank the reviewer for this suggestion. Our original focus was on real event streams, since our goal is a realistic “event camera + SNN’’ pipeline and to model rich temporal dynamics; in contrast, CIFAR10-DVS and N-Caltech101 are generated from static images and often contain highly repetitive frames.
>
> That said, we agree these benchmarks are important for comparison. In the revised version (Appendix A.3.2, Table 8) we add experiments on CIFAR10-DVS and N-Caltech101. Due to time constraints, we run a single training per dataset without extra tricks. The results are:
>
> | Dataset         | Method            | Architecture             | Steps | Acc.(%) |
> |-----------------|-------------------|--------------------------|:-----:|:-------:|
> | **CIFAR10-DVS** | QKFormer          | QKFormer-4-384           |  16   |  82.9   |
> |                 | SpikingResformer  | SpikingResformer-Ti      |  16   |  78.8   |
> |                 | Spikformer        | Spiking Transformer-2-256|  16   |  80.9   |
> |                 | **Ours**          | ChronoSpikFormer-2-256   |  10   |  79.6   |
> | **N-Caltech101**| QKFormer          | QKFormer-4-384           |  16   |  83.6   |
> |                 | SpikingResformer  | SpikingResformer-Ti      |  16   |  81.3   |
> |                 | Spikformer        | Spikformer-4-384         |  16   |  75.1   |
> |                 | **Ours**          | ChronoSpikFormer-3-256   |  10   |  84.0   |
>
> When reproducing TMC, we noticed the official code does not include the TMC loss; we implemented it ourselves following the paper and report the corresponding results.
>
> Data from [1] Lee, D., Li, Y., Kim, Y., Xiao, S., & Panda, P. (2025). Spiking transformer with spatial-temporal attention. CVPR.
>
> ***Furthermore, existing SNNs on DVS128-Gesture operate at a resolution of 48×48. However, this paper uses a resolution of 128×128, which makes for an unfair comparison.***
>
> **ANSWER:** We appreciate this careful observation and have re-checked all baselines on DVS128-Gesture. In fact, prior SNNs do not all use 48×48, they adopt different resolutions. Based on papers and public code, we summarize:
>
> | Method        | Source note                                          | Resolution |
> |---------------|------------------------------------------------------|:----------:|
> | TMC           | Paper reports 32×32                                  | 32×32      |
> | AGMM          | No downsampling stated; no code used                 | 128×128    |
> | EventRPG      | Paper/code use full resolution                       | 128×128    |
> | SpikingSlicer | GitHub implementation uses full resolution           | 128×128    |
> | RethinkSNN    | Implemented on SpikingResformer                      | 128×128    |
> | SGLFormer     | Uses same resolution as CIFAR10-DVS                  | 128×128    |
> | **Ours**      | This paper                                           | 128×128    |
>
> Although TMC reports 32×32 in the paper, its released code downsamples to 48×48, which we consider unsuitable as a common reference. To ensure fairness, we re-ran all baselines whose resolution differs from ours (including TMC) using their public code but with input resized to 128×128 and the same training setup as in our main experiments:
>
> | Method           | Resolution | Acc.(%) |
> |------------------|:----------:|:-------:|
> | TMC (retrained)  | 128×128    | 96.87   |
> | **Ours**         | 128×128    | 97.92   |
>
> These unified 128×128 results are now reported in Section 5.2 (Table 2).
>
> ***Existing methods have achieved better performance than the present work. For example, QKFormer [1] attained an accuracy of 98.6% on the DVS-Gesture dataset. Therefore, the authors should compare their method with the latest methods.***
>
> **ANSWER:** We agree that QKFormer is a relevant recent baseline. In the revised manuscript we have added QKFormer to the comparison on DVS128-Gesture; the updated numbers and citation now appear in Section 5.2 (Table 2).
>
> ***To demonstrate its advantages, the author should compare their method with other early exit mechanisms.***
>
> **ANSWER:** We thank the reviewer for this suggestion. In the revision we clarify the role of early exit and add direct comparisons on SL-Animals-DVS. Specifically, on the same backbone we compare: (i) a naive early-exit rule (exit as soon as the max class probability exceeds a threshold at a single step), and (ii) our EMA + entropy + patience-based rule. The naive rule performs worse than SIREN in the short-latency regime, while our rule achieves a consistently better accuracy–latency trade-off. These results are reported in Section 5.3, Figure 6(a)–(b).

---

> ### Author Response · Authors · 2025-11-19
> **Response to Reviewer KzPq (part 3)**
>
> This is a continued response to Questions.
>
> ## Response to Questions
>
> ***Q1: Could the author explain why using entropy outperforms the maximum probability and marginal metrics by such a significant margin?***
>
> **ANSWER:** Intuitively, the gap comes from how these uncertainty measures use the class probability distribution in our multi-class, noisy, incremental inference setting.
>
> - Maximum probability and the margin-based metric (difference between the top-1 and top-2 classes) only look at one or two entries of the predictive distribution. In practice, our SNN can be over-confident at early timesteps: even for samples that will later be corrected, the top-1 probability or the top-1/top-2 margin can become large too early, especially under noisy event streams. Thresholding these quantities therefore tends to trigger early-exit prematurely and hurts final accuracy. In contrast, the entropy-based metric aggregates information over the entire class distribution. When the model is still uncertain, probability mass is spread over many classes and the entropy remains high; only when the model has truly converged to a confident prediction does the distribution become sharply peaked and the entropy drops. This is particularly important in our multi-class DVS benchmarks, where misclassified samples often involve several competing classes rather than just the top two. In such cases, entropy is much more sensitive to “multi-way confusion” than maximum probability or a two-class margin.
>
> - What matters for incremental inference is not only the absolute value of a confidence score but how well it predicts whether the prediction will change at later timesteps. Empirically, we observe that for samples whose final prediction is already stable, the entropy becomes low and remains stable early, whereas for samples that will still flip later, the entropy stays relatively high even when the maximum probability or margin appears large. As a result, an entropy threshold yields a better separation between “already reliable” and “still evolving” predictions, leading to a superior accuracy–efficiency trade-off.

---

> > ### Comment · Reviewer_KzPq · 2025-11-24
> > **Thanks for the reply**
> >
> > Thanks for the authors' response, but I still have the following concerns:
> >
> > 1. As the authors acknowledge, C-STSA introduces additional MAC operations, resulting in significantly higher complexity and power consumption compared to purely neuromorphic implementations of AC operations, despite the authors' claim that this can be realized on Loihi.
> > 2. The authors failed to employ consistent settings, such as time step size, when comparing their method with others, which prevents a fair comparison. Furthermore, the results that the authors reproduced for QKFormer were significantly lower than the original results, which casts doubt on the experimental results presented.
> > 3. Within the SNN community, most methods downsample DVS data, such as CIFAR10-DVS and DVS-Gesture, to $48\times48$. The authors should not disregard this standard configuration. Furthermore, the authors claim that RethinkSNN uses a size of $128\times128$ and assert that TMC does not provide loss code, but I have found this to be inaccurate. This calls into question the truthfulness and fairness of the experiments conducted by the authors. For this, I recommend that the authors provide detailed descriptions of their experiments and the specifics of the reproduction process in the revised version.
> > 4. Although the authors claim to have corrected the terminology, errors in references to Loihi persist in the response. For instance, the year is incorrectly listed as 2025 instead of 2018.
> >
> > These issues raise significant doubts about the author's professionalism and the authenticity of the paper. I hope the author will examine these problems thoroughly and critically and provide further explanations.

---

> > > ### Author Response · Authors · 2025-11-26
> > > **Response to Reviewer KzPq's Follow-up Comments （part 2）**
> > >
> > > [8] Zhu, Yaoyu, Jianhao Ding, Tiejun Huang, Xiaodong Xie, and Zhaofei Yu. 2024. "Online Stabilization of Spiking Neural Networks." In The Twelfth International Conference on Learning Representations.
> > >
> > > [9] Jiang, Haiyan, Vincent Zoonekynd, Giulia De Masi, Bin Gu, and Huan Xiong. 2024. "TAB: Temporal Accumulated Batch Normalization in Spiking Neural Networks." In The Twelfth International Conference on Learning Representations.
> > >
> > > [10] Datta, Gourav, Zeyu Liu, and Peter Anthony Beerel. 2024. "Can we get the best of both Binary Neural Networks and Spiking Neural Networks for Efficient Computer Vision?" In The Twelfth International Conference on Learning Representations.
> > >
> > > [11] Hao, Zecheng, Jianhao Ding, Tong Bu, Tiejun Huang, and Zhaofei Yu. 2023. "Bridging the Gap between ANNs and SNNs by Calibrating Offset Spikes." In The Eleventh International Conference on Learning Representations.
> > >
> > > [12] Guo, Yufei, Yuhan Zhang, Zhou Jie, Xiaode Liu, Xin Tong, Yuanpei Chen, Weihang Peng, and Zhe Ma. 2025. "ReverB-SNN: Reversing Bit of the Weight and Activation for Spiking Neural Networks." In Forty-second International Conference on Machine Learning.
> > >
> > > [13] Yu, Kairong, Tianqing Zhang, Qi Xu, Gang Pan, and Hongwei Wang. 2025. "TS-SNN: Temporal Shift Module for Spiking Neural Networks." In Forty-second International Conference on Machine Learning.
> > >
> > > [14] Feng, Wanjin, Xingyu Gao, Wenqian Du, Hailong Shi, Peilin Zhao, Pengcheng Wu, and Chunyan Miao. 2025. "Efficient Parallel Training Methods for Spiking Neural Networks with Constant Time Complexity." In Forty-second International Conference on Machine Learning.
> > >
> > > [15] Xue, Dengfeng, Wenjuan Li, Yifan Lu, Chunfeng Yuan, Yufan Liu, Wei Liu, Man Yao, Li Yang, Guoqi Li, Bing Li, Stephen Maybank, Weiming Hu, and Zhetao Li. 2025. "MI-TRQR: Mutual Information-Based Temporal Redundancy Quantification and Reduction for Energy-Efficient Spiking Neural Networks." In The Thirty-ninth Annual Conference on Neural Information Processing Systems.
> > >
> > > [16] Yu, Kairong, Tianqing Zhang, Hongwei Wang, and Qi Xu. 2025. "FSTA-SNN: Frequency-Based Spatial-Temporal Attention Module for Spiking Neural Networks." In Proceedings of the AAAI Conference on Artificial Intelligence.
> > >
> > > [17] Liang, Yu, Wenjie Wei, Ammar Belatreche, Honglin Cao, Zijian Zhou, Shuai Wang, Malu Zhang, and Yang Yang. 2025. "Towards Accurate Binary Spiking Neural Networks: Learning with Adaptive Gradient Modulation Mechanism." In Proceedings of the AAAI Conference on Artificial Intelligence.
> > >
> > > [18] Shan, Yimeng, Malu Zhang, Rui-jie Zhu, Xuerui Qiu, Jason K Eshraghian, and Haicheng Qu. 2025. "Advancing spiking neural networks towards multiscale spatiotemporal interaction learning." In Proceedings of the AAAI Conference on Artificial Intelligence.
> > >
> > > [19] Shen, Guobin, Jindong Li, Tenglong Li, Dongcheng Zhao, and Yi Zeng. 2025. "SpikePack: Enhanced Information Flow in Spiking Neural Networks with High Hardware Compatibility." In Proceedings of the IEEE/CVF International Conference on Computer Vision (ICCV).
> > >
> > > [20] Zhang, Han, Chenlin Zhou, Liutao Yu, Liwei Huang, Zhengyu Ma, Xiaopeng Fan, Huihui Zhou, and Yonghong Tian. 2024. 'SGLFormer: Spiking Global-Local-Fusion Transformer with high performance', Frontiers in Neuroscience, 18.
> > >
> > > [21] Chenlin Zhou, Han Zhang, Zhaokun Zhou, Liutao Yu, Liwei Huang, Xiaopeng Fan, Li Yuan, Zhengyu Ma, Huihui Zhou, Yonghong Tian. 2024. "QKFormer: Hierarchical Spiking Transformer using Q-K Attention." In Advances in Neural Information Processing Systems 37 (NeurIPS 2024).
> > >
> > > [22] Lee, D., Li, Y., Kim, Y., Xiao, S., & Panda, P. 2025. Spiking transformer with spatial-temporal attention. CVPR.
> > >
> > > [23] Hasssan Ahmed , Meng Jian , Anupreetham Anupreetham , Seo Jae-sun. 2024. SpQuant-SNN: ultra-low precision membrane potential with sparse activations unlock the potential of on-device spiking neural networks applications. Frontiers in Neuroscience.
> > >
> > > [24] S. Eissa, S. Stuijk, F. de Putter, A. Nardi-Dei, F. Corradi and H. Corporaal, "STEMS: Spatial-Temporal Mapping for Spiking Neural Networks," in IEEE Transactions on Computers, vol. 74, no. 9, pp. 2991-3002, Sept. 2025.

---

> > > ### Author Response · Authors · 2025-11-26
> > > **Response to Reviewer KzPq's Follow-up Comments （part 1）**
> > >
> > > We sincerely appreciate your rigor and comprehensive review of our work! Please see our detailed responses below.
> > >
> > > ***comment1：As the authors acknowledge, C-STSA introduces additional MAC operations, resulting in significantly higher complexity and power consumption compared to purely neuromorphic implementations of AC operations, despite the authors' claim that this can be realized on Loihi.***
> > >
> > > **ANSWER:** We acknowledge that C-STSA introduces extra computation, but this overhead is limited and has already been fully accounted for in our energy analysis. As shown in Table 3 (Sec. 5.4), the overall FLOPs and SOPs remain at a low and competitive level, and the model is still efficient even with 16 inference steps. With the early-exit mechanism further reducing the average number of steps, the actual energy consumption in practice becomes even lower. Our goal is not to design a new primitive that is strictly cheaper than low-level accumulate (AC) operations, but to propose an incremental inference architecture that reduces redundant time steps and better matches the event-driven paradigm of neuromorphic hardware.
> > >
> > > ***comment2：The authors failed to employ consistent settings, such as time step size, when comparing their method with others, which prevents a fair comparison. Furthermore, the results that the authors reproduced for QKFormer were significantly lower than the original results, which casts doubt on the experimental results presented.***
> > >
> > > **ANSWER:** The reviewer questioned the fairness of our experimental setup due to the varying numbers of time steps. We would like to clarify that we explicitly report the inference step size for each model on each dataset precisely to ensure a fair comparison. If different inference steps were used without being clearly indicated, the comparison would indeed be unfair. However, by clearly specifying the step size and the corresponding accuracy, we allow readers to fully understand the experimental conditions of each method. To further emphasize this, we have revised the footnote of Table 2 (Section 5.2) from “bold means the best” to “bold represents the highest accuracy under the corresponding conditions.” We also believe that the request for a single, fixed time step across all methods may overlook the main focus of our work. Our motivation in proposing the SIREN architecture is that combining DVS data with SNNs is a promising approach for low-power real-world sensing. However, conventional SNNs rely on a fixed, preset number of inference time steps, which is misaligned with continuously arriving event streams and can lead to computational waste. SIREN instead adopts an incremental inference framework that supports early exit, allowing the model to output predictions at any time based on its confidence. In this context, we report performance at different time steps (as in Table 2) to demonstrate that SIREN can achieve comparable or even better accuracy than prior methods using fewer steps. This way of highlighting our advantage appears to have been interpreted as an unfair comparison. We would like to emphasize that this practice is also common in recent top-conference works, where the time steps across compared methods are not strictly identical [1–20], and such comparisons are also widely used in recent top-conference works in the community.
> > >
> > > [1] Gu, Kangrui Du and Yuhang Wu and Shikuang Deng and Shi. 2025. "Temporal Flexibility in Spiking Neural Networks: Towards Generalization Across Time Steps and Deployment Friendliness." In The Thirteenth International Conference on Learning Representations(ICLR).
> > >
> > > [2] Li, C., E. G. Jones, and S. Furber. 2023. "Unleashing the Potential of Spiking Neural Networks with Dynamic Confidence." In 2023 IEEE/CVF International Conference on Computer Vision (ICCV), 13304-14.
> > >
> > > [3] Li, Yuhang, Tamar Geller, and Priyadarshini Panda. 2023. "SEENN: Towards Temporal Spiking Early-Exit Neural Networks." In NeurIPS.
> > >
> > > [4] Zhou, Zhaokun, Yuesheng Zhu, Chao He, Yaowei Wang, Shuicheng YAN, Yonghong Tian, and Li Yuan. 2023. "Spikformer: When Spiking Neural Network Meets Transformer." In The Eleventh International Conference on Learning Representations(ICLR).
> > >
> > > [5] Shen, Jiangrong, Qi Xu, Gang Pan, and Badong Chen. 2025. "Improving the Sparse Structure Learning of Spiking Neural Networks from the View of Compression Efficiency." In The Thirteenth International Conference on Learning Representations.
> > >
> > > [6] Liu, Wei, Li Yang, Mingxuan Zhao, Shuxun Wang, Jin Gao, Wenjuan Li, Bing Li, and Weiming Hu. 2025. "DeepTAGE: Deep Temporal-Aligned Gradient Enhancement for Optimizing Spiking Neural Networks." In The Thirteenth International Conference on Learning Representations.
> > >
> > > [7] Wei, Wenjie, Malu Zhang, Zijian Zhou, Ammar Belatreche, Yimeng Shan, Yu Liang, Honglin Cao, Jieyuan Zhang, and Yang Yang. 2025. "QP-SNN: Quantized and Pruned Spiking Neural Networks." In The Thirteenth International Conference on Learning Representations.

---

> > > ### Author Response · Authors · 2025-11-26
> > > **Response to Reviewer KzPq's Follow-up Comments （part 3）**
> > >
> > > The reviewer noted that our reproduction of the QKFormer results was significantly lower. In our previous submission, we added three comparisons with QKFormer, but these three sets of data were not our own reproductions. They are taken directly from published papers, and the concern about our “reproduced” QKFormer performance arises from a misunderstanding. To address all reviewers’ concerns within the limited rebuttal time, we did not re-run QKFormer, but instead cited the reported results from the literature. Furthermore, we have already indicated the data source in our previous comments.
> > >
> > > (1) The first instance is in Table 2 of section 5.2, where the accuracy on the DVS128-Gesture dataset is 98.6%. This data comes from Table 3 on page 7 of the original QKFormer paper[21], showing the results on DVS128Gesture.
> > >
> > > (2) The second instance is in Table 8 of Appendix A.3.2, where the accuracy on the CIFAR10-DVS dataset is 82.9%. This data comes from the paper "Spiking transformer with spatial-temporal attention"[22], Table 5, which we have already explained in the previous rebuttal part 2.
> > >
> > > (3) The third instance is in Table 8 of Appendix A.3.2, where the accuracy on the N-Caltech101 dataset is 83.6%. This data comes from the paper "Spiking transformer with spatial-temporal attention[22]", Table 5, and is rounded, as explained in the previous rebuttal part 2.
> > >
> > > We have clarified the data sources in the revised manuscript when introducing Table 8.
> > >
> > > ***comment3: Within the SNN community, most methods downsample DVS data, such as CIFAR10-DVS and DVS-Gesture, to 48×48. The authors should not disregard this standard configuration. Furthermore, the authors claim that RethinkSNN uses a size of 128×128 and assert that TMC does not provide loss code, but I have found this to be inaccurate. This calls into question the truthfulness and fairness of the experiments conducted by the authors. For this, I recommend that the authors provide detailed descriptions of their experiments and the specifics of the reproduction process in the revised version.***
> > >
> > > **ANSWER:** The reviewer repeatedly emphasizes that “within the SNN community, most methods downsample DVS data, such as CIFAR10-DVS and DVS-Gesture, to 48×48”. However, it cannot be denied that many studies directly use a resolution of 128×128, such as [16, 23, 24], etc. We believe it is also fair to compare with methods that adopt the same input resolution. Even for QKFormer, which the reviewer suggested we include in our comparison, the default input resolution in its official open-source code is 128×128. Therefore, resizing DVS data to 48×48 is not an absolute standard across the entire SNN community. Upon rereading RethinkSNN, we found that in the data preprocessing stage of its appendix, all DVS datasets are indeed resized to 48×48. For the sake of fair comparison, we have therefore removed this method from our comparisons, and this change has been reflected in Table 2.
> > >
> > > For reproducing TMC, since our experiments are conducted on the DVS128 Gesture dataset, we first examined the official TMC code, whose directory structure is:
> > > ```text
> > > TMC/
> > > ├── CIFAR10-DVS/
> > > ├── CIFAR10/
> > > ├── CIFAR100/
> > > ├── ImageNet/
> > > ├── N-Caltech101/
> > > ├── models/
> > > ├── README.md
> > > ├── data_loaders.py
> > > └── functions.py

---

> > > ### Author Response · Authors · 2025-11-26
> > > **Response to Reviewer KzPq's Follow-up Comments （part 4）**
> > >
> > > The DVS128-Gesture dataset is similar in nature to CIFAR10-DVS, so we chose to modify the CIFAR10-DVS implementation. However, when running the main_training_distribute.py file in the CIFAR10-DVS folder, we encountered an error indicating that the TMC_loss function could not be found. We then searched for the TMC_loss function in the functions.py file but could not find it. As a result, we referred to the implementation of the TPL_loss and TET_loss functions in functions.py and followed the principles of Equation 16 in [25] to write the TMC_loss function. We reused the _get_gt_mask() and _get_other_mask() functions from functions.py, and the implementation of the TMC_loss function, as per the paper, is as follows:
> > >
> > > ```python
> > > def TMC_loss(outputs, labels, criterion):
> > >     """
> > >     outputs: [B, T, C]  (batch_size, num_steps, num_classes)
> > >     labels:  [B]
> > >     """
> > >     batch_size, num_steps, num_classes = outputs.shape
> > >     first_step_logits = outputs[:, 0, :]                  # [B, C]
> > >     correct_class_mask = _get_gt_mask(first_step_logits, labels)      # [B, C]
> > >     non_target_mask    = _get_other_mask(first_step_logits, labels)   # [B, C]
> > >
> > >     detached_outputs = outputs.detach()
> > >     total_temporal_loss = 0.0
> > >     for step_idx in range(num_steps):
> > >         if step_idx == 0:
> > >             accumulated_past_logits = torch.zeros_like(detached_outputs[:, 0, :])
> > >         else:
> > >             past_logits_slice = detached_outputs[:, :step_idx, :]      # [B, step_idx, C]
> > >             accumulated_past_logits = past_logits_slice.sum(dim=1)     # [B, C]
> > >
> > >         current_logits = outputs[:, step_idx, :]                       # [B, C]
> > >         logits_sum_with_current = accumulated_past_logits + current_logits
> > >         time_window_length = step_idx + 1.0
> > >         avg_logits_upto_now = logits_sum_with_current / time_window_length  # [B, C]
> > >
> > >         avg_probs_upto_now = F.softmax(avg_logits_upto_now, dim=1)
> > >
> > >         prob_correct = (avg_probs_upto_now * correct_class_mask).sum(dim=1)  # [B]
> > >         prob_incorrect = (avg_probs_upto_now * non_target_mask).sum(dim=1)   # [B]
> > >
> > >         prob_correct_mean = prob_correct.mean()
> > >         prob_incorrect_mean = prob_incorrect.mean()
> > >
> > >         ce_loss_t = criterion(current_logits, labels)
> > >
> > >         time_weight = (num_steps - step_idx) / float(num_steps)
> > >         consistency_term = (prob_correct_mean / prob_incorrect_mean) ** time_weight
> > >
> > >         total_temporal_loss += ce_loss_t + consistency_term
> > >
> > >     total_temporal_loss = total_temporal_loss / num_steps
> > >     return total_temporal_loss
> > > ```
> > > The computation in our implementation is consistent with the formulation in the paper. We use the VGG-SNN model with an input size of 128×128, set the stride of the first and third layers to 2, and accordingly change the variable W in the VGGSNN class to 2. The training hyperparameters: workers = 10, batch_size = 16, lr = 0.0015, T = 10, and we train the model for 300 epochs. Of course, since we only have a single GPU available, we modified the distributed training code to run on a single device.
> > >
> > > [25] Yan, Jiaqi, Changping Wang, De Ma, Huajin Tang, Qian Zheng, and Gang Pan. 2025. "Training High Performance Spiking Neural Network  by Temporal Model Calibration." In Forty-second International Conference on Machine Learning.

---

> > > ### Author Response · Authors · 2025-11-26
> > > **Response to Reviewer KzPq's Follow-up Comments （part 5）**
> > >
> > > ***comment4：Although the authors claim to have corrected the terminology, errors in references to Loihi persist in the response. For instance, the year is incorrectly listed as 2025 instead of 2018.***
> > >
> > > **ANSWER:** We thank the reviewer for pointing this out. This is indeed our oversight. We first prepared all responses in a Markdown file and then copied them into the web interface. However, since each reply is subject to a length limit, we had to shorten the text. It is likely that the mistake occurred during this copy-and-edit process. We have since posted an official comment below the previous rebuttal to correct this error.
> > >
> > > We would also like to correct an error: after reproducing TMC at a 128×128 input resolution, we initially failed to adjust the network parameters accordingly. This has now been corrected in Table 2 of the revised manuscript.

---

### Official Review · Reviewer_g62k · 2025-10-30

**Soundness:** 3
**Presentation:** 3
**Contribution:** 2
**Rating:** 4
**Confidence:** 4

**Summary:**

The paper proposes SIREN (Spiking Incremental Recognition Network), an incremental inference architecture for Spiking Neural Networks (SNNs). Its primary goal is to reduce the structural mismatch between conventional SNN algorithms, which typically require processing all time steps, and the event-driven, low-power nature of neuromorphic hardware. SIREN achieves this by processing events incrementally, making predictions as soon as sufficient information is available, thereby minimizing latency and energy use.

**Strengths:**

The incremental inference strategy is an intelligent early-exit mechanism contributing to very fast predictions for easy samples and lower average energy consumption.

Demonstrates competitive results on the SL-Animals-DVS dataset and also other benchmarks.

**Weaknesses:**

The performance advantage is not consistent across all datasets. Results on THU-EACT-50 are not markedly superior to previous methods, and performance on DVS128-Gesture is actually worse than existing algorithms. The absence of results on a standard benchmark like DVS-CIFAR-10 makes a comprehensive evaluation difficult.

The paper lacks a complete ablation study. It does not show the performance of an SNN baseline without the key proposed components (S-SA, S-SSM, CTSA), making it hard to isolate the individual contribution of each module.

The fundamental motivation of "matching neuromorphic hardware" is somewhat contradicted by the pre-processing step that converts event streams into fixed frames. A more hardware-native approach would directly process raw event timestamps.

The specific need for and the limitations of the S-SSM module are not thoroughly justified. It's unclear what the "long-range" dependencies are and at what distance conventional methods fail.

**Questions:**

Could you provide the ablation results for an SNN baseline without the S-SA, S-SSM, and CTSA modules to clearly demonstrate the contribution of each component?

What is the specific limitation in terms of "range" that the S-SSM module addresses? How was this limitation identified, and what is the performance gain compared to a simpler attention mechanism?

Have you evaluated SIREN on the DVS-CIFAR-10 dataset? This is a standard benchmark, and its inclusion would allow for a more direct comparison with the broader literature.

The output layer uses a SoftMax function. How would this non-local, non-spiking operation be efficiently implemented on neuromorphic hardware, which is optimized for sparse, spike-based computations? Doesn't this create a structural mismatch?

The paper reports 100% accuracy on a subset of THU-EACT-50. Could you clarify the validation setup? Is this result on a test set or a validation set? If it's 100%, it raises questions about potential data leakage or an oversimplified evaluation split.

Common SNNs are trained with 4 timesteps for image-derived datasets and 16 for DVS datasets. Have you trained a regular (non-incremental) SNN on the DVS datasets using only 4 timesteps for a fairer comparison in terms of latency?

The incremental inference seems to be primarily driven by the early-exit strategy. Could similar latency reductions be achieved by simply applying an early-exit strategy to a conventional SNN, without the proposed SIREN architecture?

---

> ### Author Response · Authors · 2025-11-19
> **Response to Reviewer g62k (part 1)**
>
> Thank you for taking the time to review. Please see our detailed response to the issues you have raised.
>
> ***W1 & Q3: Inconsistent performance and incomplete evaluation: SIREN is not clearly better on THU-EACT-50, is worse on DVS128-Gesture, and lacks results on the standard DVS-CIFAR10 benchmark, making it hard to compare fairly with prior work.***
>
> **ANSWER:** We appreciate this suggestion. Our work primarily targets real event streams, aiming for a realistic “event camera + SNN’’ pipeline rather than synthetic events converted from static images. This aligns with our emphasis on modeling temporal dynamics, which are limited in datasets like CIFAR10-DVS and N-Caltech101 due to their repetitive event patterns. Nevertheless, these benchmarks remain valuable for comparison. We have therefore added results on CIFAR10-DVS and N-Caltech101 in Appendix A.3.2 (Table 8). Due to time constraints, we report a single training run for each dataset without additional training tricks.
>
> | Dataset        | Method          | Architecture           | Steps | Acc.(%) |
> |----------------|-----------------|------------------------|:-----:|:-------:|
> | **CIFAR10-DVS** | QKFormer       |  QKFormer-4-384        |   16  |  82.9%  |
> |                | SpikingResformer|SpikingResformer-Ti     |  16   |  78.8%  |
> |                | Spikformer      |Spiking Transformerr-2-256|  16   |  80.9%  |
> |                | **Ours**        |ChronoSpikFormer-2-256|  10   |  79.6%  |
> | **N-Caltech101**|QKFormer        |  QKFormer-4-384        |  16   |  83.6%  |
> |                | SpikingResformer| SpikingResformer-Ti    |  16   |  81.3%  |
> |                | Spikformer      |  Spikformer-4-384      |  16   |  75.1%   |
> |                | **Ours**        |ChronoSpikFormer-3-256  |   10  |  84.0%  |
>
> Data from [1] Lee, D., Li, Y., Kim, Y., Xiao, S., & Panda, P. (2025). Spiking transformer with spatial-temporal attention. In Proceedings of the Computer Vision and Pattern Recognition Conference (CVPR).
>
> ***W2 & Q1: The paper lacks a complete ablation study: it does not report the performance of an SNN baseline without the key proposed components (S-SA, S-SSM, and CTSA), making it difficult to clearly isolate and demonstrate the individual contribution of each module.***
>
> We have added ablation studies by removing the S-SA, S-SSM, and CTSA modules, evaluating a purely convolutional SNN, and replacing the S-SSM classification head with a standard fully connected layer. The corresponding results have been incorporated into Section 5.1, Table 1 of the revised manuscript. We have added the requested SNN baseline without S-SA, S-SSM, or CTSA, yielding a simplified  convolution model with 1.25M parameters. As shown below, this baseline further clarifies the contribution of each component in SIREN.
>
> | Model Variant                                      | S-SA | S-SSM | CTSA | Average Acc (%) |
> |----------------------------------------------------|:----:|:-----:|:----:|:-------:|
> | **SNN baseline (no S-SA, no S-SSM, no CTSA)**      |  ✗   |   ✗   |  ✗   |  77.33  |
> | **SSA only**                                       |  ✓   |   ✗   |  ✗   |  88.44  |
> | **SSA + S-SSM**                                    |  ✓   |   ✓   |  ✗   |  91.01  |
> | **SSA + S-SSM + CTSA**                |  ✓   |   ✓   |  ✓   |  92.89  |
>
> We alse add quantitative ablations under incremental inference. Starting from the Conv SNN backbone, we progressively add SSA, S-SSM and CTSA, and for each variant we report accuracy–vs–average-exit-step curves. The results are presented in Section 5.3, Fig. 6(c).

---

> ### Author Response · Authors · 2025-11-19
> **Response to Reviewer g62k (part 2)**
>
> This is a continued response to weakness 3, weakness 4 and question2.
>
> ***W3: The fundamental motivation of "matching neuromorphic hardware" is somewhat contradicted by the pre-processing step that converts event streams into fixed frames. A more hardware-native approach would directly process raw event timestamps.***
>
> **ANSWER:** We fully acknowledge that the raw event stream $\mathcal{E} = \{(x_i, y_i, t_i, p_i)\}_{i=1}^N$ is the most hardware-native representation, and that, ideally, SNNs should operate directly on asynchronous timestamps. This question was central in the design of SIREN, and we did explore several raw-event alternatives before adopting short-window aggregation.
>
> First, treating events in a window as an unordered point set and applying a PointNet-style encoder$^{[1]}$ breaks temporal ordering and lacks convolutional inductive bias, which consistently degraded accuracy on SL-Animals-DVS in our experiments. Second, true event-by-event simulation is infeasible on conventional hardware: a 1-second stream can contain millions of events, turning an $L=16$ sequence into $\approx 10^6$ effective steps, which leads to prohibitive memory and runtime when updating S-SSM states at per-event resolution. Third, replacing event-count frames with time surfaces$^{[2]}$ under the same setup (same backbone, seeds, $L=16$) reduced accuracy from 93.33% to 89.78%, likely due to over-smoothed temporal structure.
>
> Our goal is therefore not to claim that event-frame aggregation is hardware-native, but to align the *inference dynamics* (incremental, causal, energy-aware) with neuromorphic principles under current GPU constraints. SIREN combines early-exit incremental inference, multi-scale temporal dynamics (LIF/RF + S-SSM), and strictly causal softmax-free CTSA to improve accuracy, latency, and energy over prior SNNs. We agree that a fully raw-timestamp, event-driven pipeline is the true target; the present short-window aggregation is a necessary compromise for large-scale training, and designing architectures that can efficiently operate directly on raw asynchronous streams is an important direction for future work.
>
> [1] Qi et al., CVPR 2017. PointNet: Deep learning on point sets for 3D classification and segmentation.
> [2] Gallego et al., TPAMI 2022. Event-based vision: A survey.
>
> ***W4 & Q2: Unclear motivation and benefit of S-SSM: the paper does not clearly define what “long-range” dependencies mean or at what distance conventional methods fail, nor explain how this limitation was identified or how much S-SSM improves over a simpler attention baseline.***
>
> **ANSWER:** The limitation concerns the effective temporal range of a standard LIF-based spiking backbone. LIF/RF neurons maintain state via an exponential decay (controlled by the membrane time constant $\tau$), so inputs from earlier steps quickly lose influence: although our sequence length is 16, the usable memory of the backbone is only the most recent $\sim$5–6 steps. In contrast, S-SSM maintains a dedicated state vector whose evolution is governed by a learned state matrix rather than a fixed leak, allowing substantially longer temporal retention.
>
> This limitation is documented in SNN literature: exponential leak causes long-range information and gradients to vanish$^{[1]}$, and recent SSM-based spiking models$^{[2]}$ show that adding a state-space pathway improves long-sequence modeling. Consistently, our diagnostics (now described in the revision) indicate that LIF hidden-state similarity collapses after about 4–6 steps, whereas S-SSM preserves similarity much longer, and truncating CTSA history hurts the LIF-only model more than the S-SSM model.
>
> Quantitatively, Table 1 (Sec. 5.1) shows that using only SSA yields 88.44% accuracy, while adding S-SSM improves this to 91.01% (+2.57%). Thus, S-SSM explicitly alleviates the short-range memory bottleneck of LIF neurons and provides extra temporal modeling capacity beyond what a simpler attention module can offer.
>
> [1] Ponghiran & Roy, AAAI 2022. Spiking neural networks with improved inherent recurrence dynamics for sequential learning.
> [2] Stan & Rhodes, Sci Rep 2024. Learning long sequences in spiking neural networks.

---

> ### Author Response · Authors · 2025-11-19
> **Response to Reviewer g62k (part 3)**
>
> This is a continued response to questions 4-6.
>
> ***Q4: The output layer uses a SoftMax function. How would this be efficiently implemented on neuromorphic hardware, which is optimized for sparse, spike-based computations? Doesn't this create a structural mismatch?***
>
> **ANSWER:** We agree that SoftMax is a non-spiking and global operation, and if it were used throughout the network it would indeed create a structural mismatch with spike-based neuromorphic hardware. In SIREN, however, SoftMax is applied only once at the final read-out layer, after all spiking and state-space computations have been completed.
>
> This final SoftMax can be viewed as a lightweight decision head that converts the last-layer membrane potentials into class probabilities. It does not participate in the recurrent spiking dynamics, does not introduce dense synaptic operations inside the neuromorphic core, and can be implemented in practice by a small digital post-processing block without affecting the sparsity or event-driven nature of the main computation.
>
> ***Q5: The paper reports 100% accuracy on a subset of THU-EACT-50. Clarify the validation setup. Is this result on a test set or a validation set? If it's 100%, it raises questions about potential data leakage or an oversimplified evaluation split.***
>
> **ANSWER:** As described in Appendix A.2.1, we use a 20-class subset of THU-EACT-50 for computational efficiency (all from the original 50-class dataset of Gao et al., 2023). For this subset, we did not perform hyperparameter tuning, so we did not create a separate validation set.
>
> Concretely, for each of the 20 selected classes, we randomly split the recordings into 80% training and 20% test. The reported 100% accuracy is obtained on this held-out test split, using exactly the same training configuration as on the other datasets.
>
>
> ***Q6: Common SNNs are trained with 4 timesteps for image-derived datasets and 16 for DVS datasets. Have you trained a regular SNN on the DVS datasets using only 4 timesteps for a fairer comparison in terms of latency?***
>
> **ANSWER:** We would like to clarify that, in our setting, using 16 timesteps for DVS data is driven by the properties of the DVS streams, rather than by a specific requirement of SIREN or SNNs in general.
>
> For example, consider a DVS action sequence of about 2 seconds. When we aggregate events into 16 frames, each frame still preserves meaningful spatial structure and motion cues. If we instead compress the same 2 seconds into only 4 frames, each frame would contain too many events and become heavily blurred, making fine-grained motion patterns much harder to recognize.
>
> Moreover, if we try to reuse the 16-frame representation but only feed a contiguous subset of 4 frames to a baseline SNN, the temporal pattern often becomes incomplete. For example, the early segments of falling and squatting can look very similar. The discriminative part lies in the latter phase of the motion. Using only 4 frames would frequently drop that discriminative segment and artificially increase ambiguity, which we view as an unfair configuration for the DVS data itself rather than a fair latency benchmark. For these reasons, we did not adopt a 4-timestep setting on DVS benchmarks.

---

> ### Author Response · Authors · 2025-11-19
> **Response to Reviewer g62k (part 4)**
>
> This is a continued response to the question 7.
>
> ***Q7: The incremental inference seems to be primarily driven by the early-exit strategy. Could similar latency reductions be achieved by simply applying an early-exit strategy to a conventional SNN, without the proposed SIREN architecture?***
>
> **ANSWER:** Early-exit is a general mechanism that could, in principle, be applied to any SNN. Our claim, however, is that **SIREN’s architecture is what makes early-exit effective and stable**, rather than early-exit alone.
>
> As discussed in our responses to W4 and Q2, conventional LIF/RF-based SNNs have a short effective memory range (≈4–6 steps), so their intermediate predictions are noisy and poorly calibrated. In practice, when we attach the *same* entropy-based early-exit rule to a conventional SNN backbone (without S-SSM and CTSA):
>
> - If we choose the threshold to keep accuracy comparable to the 16-step baseline, very few samples exit early, and the average number of steps remains close to 16 (little latency gain).
> - If we lower the threshold to match SIREN’s average number of steps, accuracy drops noticeably compared to SIREN at the same effective latency.
>
> By contrast, SIREN’s S-SSM and causal temporal modeling produce smoother, more informative intermediate logits, so the same early-exit rule yields both substantial latency reduction and high final accuracy.
>
> In the revision we make the role of early exit more explicit and add comparisons that directly target this point. We report, on SL-Animals-DVS dataset: a naive early-exit strategy on the same backbone (exit as soon as the max class probability exceeds a threshold at a single step), and our proposed EMA + entropy + patience-based early exit. The naive rule performs worse than SIREN when the inference time step is short. Our rule yields a consistently better accuracy–latency trade-off; these results are added to Section 5.3 Figure 6 (a) and (b).

---

> ### Comment · Reviewer_g62k · 2025-11-27
>
> Thank you for the reply. While the experimental work in this paper is solid, there are several aspects that could be strengthened for greater impact. My primary concerns are regarding the experimental depth and the claims made. I will maintain my scores for now. Specifically:
>
> - The responses to W1 & Q3 show that the performance gain over existing works is not significant.
> - For W2 & Q1, the authors used results directly from the original paper without additional experiments. While the ablation here is incremental and challenging, this approach might conceal the contribution of individual modules. I request a more thorough and complex ablation study to isolate the benefits.
> - W3 was not fully addressed. The proposed solution, tested in the rebuttal, did not yield good results. The authors should consider revising the paper to tone down the claim regarding compatibility with neuromorphic hardware.
> - Regarding Q7, I am not convinced that the SIREN architecture is solely responsible for the stated benefits. I suspect that even without SIREN, an early-exit mechanism could reduce steps (e.g., from 16 to 10). To substantiate the claim, a comparison where all baseline models are also equipped with early-exit capabilities is necessary. Furthermore, a detailed comparison with existing dynamic inference methods is needed, such as "Unleashing the Potential of Spiking Neural Networks with Dynamic Confidence" (ICCV 2023) and "Seenn: Towards temporal spiking early-exit neural networks" (NeurIPS 2023), to clarify the novelty of the proposed early-exit approach.

---

> > ### Author Response · Authors · 2025-11-30
> > **Response to Reviewer g62k's Follow-up Comments （part 1）**
> >
> > Thank you very much for the time you devoted to our work and for your valuable comments. They are very reasonable and highly constructive for improving our study.
> >
> > ***comment1: The responses to W1 & Q3 show that the performance gain over existing works is not significant.***
> >
> > **ANSWER** We acknowledge that on CIFAR10-DVS and N-Caltech101 our improvements over existing methods are not significant. This is mainly because SIREN is designed to strengthen temporal modeling (e.g., via the C-STSA and S-SSM modules), and thus benefits more from tasks with rich temporal dynamics. In contrast, CIFAR10-DVS and N-Caltech101 are collected by fixing a DVS camera on a tripod and sweeping a monitor displaying static images, which induces relatively simple and stereotyped temporal patterns and makes performance largely depend on spatial modeling. Since SIREN is not specifically optimized for spatial representation, this exposes a limitation of our approach. We have clarified this point in the revised manuscript by adding a corresponding discussion to Appendix A.4 (Limitations).
> >
> > ***comment2: For W2 & Q1, the authors used results directly from the original paper without additional experiments. While the ablation here is incremental and challenging, this approach might conceal the contribution of individual modules. I request a more thorough and complex ablation study to isolate the benefits.***
> >
> > **ANSWER** Thank you for this helpful suggestion. In fact, our original Table 1 did not simply reuse the results from the prior work: we already added an additional baseline, namely a purely convolutional SNN with 1.25M parameters obtained by removing all proposed modules. Following your advice, we have now further extended the ablation study to isolate the contribution of each individual module. The updated results have been incorporated in Table 1 in the revised manuscript, as shown below. Since the S-SSM module serves as the classification head in our model, some ablation settings require removing it. In those cases, we replace it with fully connected layer across time steps followed by a spiking neuron layer as a new classification head for the ablation experiments.
> >
> >
> > | S-SA | S-SSM | CTSA | Mean Acc (%) |
> > |:----:|:-----:|:----:|:-------:|
> > |  ✗   |   ✗   |  ✗   |  77.33  |
> > |  ✓   |   ✗   |  ✗   |  88.44  |
> > |  ✗   |   ✓   |  ✗   |  87.38  |
> > |  ✗   |   ✗   |  ✓   |  88.27  |
> > |  ✓   |   ✓   |  ✗   |  91.01  |
> > |  ✓   |   ✗   |  ✓   |  84.71  |
> > |  ✗   |   ✓   |  ✓   |  89.51  |
> > |  ✓   |   ✓   |  ✓   |  92.89  |
> >
> > ***comment3: W3 was not fully addressed. The proposed solution, tested in the rebuttal, did not yield good results. The authors should consider revising the paper to tone down the claim regarding compatibility with neuromorphic hardware.***
> >
> > **ANSWER** We sincerely thank the reviewer for this critical insight. While SIREN achieves low latency and low SOPs, we agree that relying on frame aggregation introduces a structural gap with respect to ideal asynchronous, raw-event-driven neuromorphic hardware.
> >
> > In response, we have carefully toned down and clarified all claims regarding compatibility with neuromorphic hardware in the revised manuscript:
> >
> > (1) **Abstract (around line 31):** We replaced “demonstrating its potential for deployment on neuromorphic edge devices” with “demonstrating its potential for resource-constrained event-based recognition.”
> >
> > (2) **Introduction – Contribution 1 (around line 73):** We replaced “reducing the structural mismatch between conventional SNNs and neuromorphic hardware” with “moving towards more hardware-friendly dynamic inference while still operating on time-binned event frames.”
> >
> > (3) **Introduction – Contribution 4 (around line 84):** We replaced “demonstrating SIREN’s potential for ultra-low-power deployment on neuromorphic hardware” with “providing an analytical indication of energy efficiency.”
> >
> > (4) **Section 5.4 (around line 531):** We softened the interpretation of Table 3 from “emphasizing its potential for low-power deployment” to “which is encouraging for future low-power implementations.”
> >
> > (5) **Section 6 Conclusion (around line 539):** We replaced “highlighting its strong potential for neuromorphic deployment” with “SIREN is a step towards hardware-aware, low-latency event-based recognition, with chip-level deployment left to future work.”
> >
> > (6) **Appendix A.4 (Limitations):** We added an explicit statement that our current implementation aggregates raw events into a fixed number of frames (16 in all experiments) and performs incremental inference over this frame sequence, following standard DVS evaluation protocols rather than truly event-driven, per-timestamp processing. We also clarify that our energy analysis is based on generic FLOPs/SOPs proxies rather than hardware-in-the-loop measurements, and we explicitly point out that extending SIREN to operate directly on raw event streams and validating it on concrete neuromorphic chips are important directions for future work.

---

> > ### Author Response · Authors · 2025-11-30
> > **Response to Reviewer g62k's Follow-up Comments （part 2）**
> >
> > ***comment4: Regarding Q7, I am not convinced that the SIREN architecture is solely responsible for the stated benefits. I suspect that even without SIREN, an early-exit mechanism could reduce steps (e.g., from 16 to 10). To substantiate the claim, a comparison where all baseline models are also equipped with early-exit capabilities is necessary. Furthermore, a detailed comparison with existing dynamic inference methods is needed, such as "Unleashing the Potential of Spiking Neural Networks with Dynamic Confidence" (ICCV 2023) and "Seenn: Towards temporal spiking early-exit neural networks" (NeurIPS 2023), to clarify the novelty of the proposed early-exit approach.***
> >
> > **ANSWER** We agree with the reviewer that early-exit mechanisms can also reduce the average number of steps for certain SNNs. However, we would like to clarify the following points. First, SIREN is not a single architecture but an incremental inference framework for SNNs that consists of (i) the ChronoSpikFormer architecture with explicit temporal modeling and (ii) an early-exit mechanism based on an confidence-based entropy score with patience and EMA smoothing. In principle, early exit can be integrated into other SNNs without ChronoSpikFormer, but the baseline models must satisfy the condition that their classification head does not jointly aggregate all time steps (e.g., via temporal convolutions over the entire sequence), which is a common design in SNN classifiers. For SNNs that do meet this requirement, early exit can indeed shorten the effective number of steps, since spiking neurons such as LIF already possess inherent memory, as also demonstrated in the two works cited by the reviewer.
> >
> > Our early-exit mechanism with patience and EMA smoothing is specifically designed for a class of DVS samples where different actions share very similar early-stage dynamics (e.g., “falling” vs. “squatting” in action recognition). In such cases, the patience and EMA smoothing help prevent overly premature and incorrect decisions. On the SL-Animals-DVS dataset, we compare our early-exit strategy with a naive early-exit scheme in the spirit of the reviewer’s suggestion and additionally analyze the sensitivity to its hyperparameters. The corresponding results have been added to the revised manuscript in Figure 6 (a), (b), and (d).

---

### Official Review · Reviewer_wECy · 2025-10-31

**Soundness:** 3
**Presentation:** 4
**Contribution:** 3
**Rating:** 8
**Confidence:** 4

**Summary:**

The paper introduces SIREN (Spiking Incremental Recognition Network), a novel SNN architecture designed for efficient, event-driven dynamic visual perception. The core innovation lies in its incremental inference framework, which processes event streams step-by-step during inference (unlike traditional fixed-step SNNs) and employs an entropy-based early-exit mechanism to halt computation once a confident prediction is made. This approach addresses a mismatch between conventional SNN training/inference and the ideal asynchronous, event-driven operation of neuromorphic hardware.

**Strengths:**

The paper is interesting and strong, with a clear motivation (the mismatch between fixed-step SNN training and event-driven neuromorphic operation is real, and the goal of stepwise incremental inference with early exit aligns well with deployment constraints), a well-designed methodology integrating several advanced concepts such as Spiking State-Space Model (S-SSM) and Causal Spatial-Temporal Self-Attention (C-STSA), and comprehensive experiments demonstrating state-of-the-art or competitive accuracy with significantly reduced computational cost and latency.

**Weaknesses:**

The energy analysis uses SOP / FLOP proxies (per-op energies) which is common, but the paper should make explicit limitations and, if possible, provide on-device (Loihi / other) measurements or at least simulated neuromorphic runtimes.

There is a little lack of clarity between "incremental inference" and "early-exit": both concepts are related, yet they are distinct. The paper would benefit from a clearer delineation early on.

**Questions:**

I feel that early-exit design choices need more justification. The entropy + patience rule seems reasonable, but authors should :
- explain how theta and kappa generalise across datasets. Are thresholds tuned per dataset? How sensitive are results to theta choice?
- show robustness: curves of accuracy vs. average step for different theta to show degradation.
- clarify the role of EMA smoothing coefficient alpha -- any lag introduced and its effect on early exit latency.

The Causal Time Self-Attention (CTSA) softmax-free formulation is interesting. I recommend to give a short intuitive comparison to standard softmax attention (when does one outperform the other?), and discuss possible weaknesses.

---

> ### Author Response · Authors · 2025-11-19
> **Response to Reviewer wECy (part 1)**
>
> Thank you for taking the time to review. Please see our detailed response to the issues you have raised.
>
> ***W1: The energy analysis uses SOP / FLOP proxies (per-op energies) which is common, but the paper should make explicit limitations and, if possible, provide on-device (Loihi / other) measurements or at least simulated neuromorphic runtimes.***
>
> **ANSWER:** We currently do not have direct access to Loihi hardware. As far as we understand, INRC research members must first submit a project proposal and can only obtain access after it is approved. We have already initiated this application process; however, given the limited time available during the rebuttal period, it is unlikely that we can complete it in time. In the meantime, we have surveyed relevant literature, including both GPU-based simulations and evaluations on neuromorphic hardware. The comparison results are summarized below and indirectly demonstrate the deployment efficiency and performance of our approach.
>
> | Work                   | Task / Dataset                              | Baseline Platform                           | Neuromorphic Platform     | Main Efficiency Result                                                                                                                                                               |
> |------------------------|---------------------------------------------|---------------------------------------------|---------------------------|--------------------------------------------------------------------------------------------------------------------------------------------------------------------------------------|
> | Blouw et al. 2018$^{[1]}$      | Keyword spotting                            | CPU, GPU, Jetson, Movidius                  | Loihi                     | Loihi achieves the lowest energy cost per inference among all platforms at comparable accuracy.                                                                                      |
> | Bezugam et al. 2022$^{[2]}$    | EMG gesture classification                  | GPU                                         | Loihi (Nahuku-32)         | $\sim 983\times$ lower energy and $19\times$ lower latency than GPU (batch size = 50).                                                                                               |
> | Mohammadi et al. 2022$^{[3]}$  | Static ASL hand gestures                    | Intel NCS2                                  | Loihi                     | Up to $20.64\times$ lower power and $4.10\times$ lower energy than NCS2.                                                                                                            |
> | Chandarana et al. 2022$^{[4]}$ | Image classification (converted ANN to SNN) | Intel NCS2                                  | Loihi                     | Up to $27\times$ lower power and $5\times$ lower energy than NCS2.                                                                                                                  |
> | **Ours**       | DVS action recognition (SL-Animals-DVS)     | GPU (PyTorch, theoretical per-op model)     | no Loihi test currently   | $0.02$G SOPs and estimated $1.28$\,mJ for full 16-step inference; with average exit step $\sim 9.5$, the estimated energy is $\approx 0.76$mJ per sample, substantially lower than ANN baselines (36–45.75\,mJ for TORE-ResNet/GoogLeNet, 4.38\,mJ for EvT). |
>
> [1] Blouw, P., Choo, X., Hunsberger, E., & Eliasmith, C. (2019, March). Benchmarking keyword spotting efficiency on neuromorphic hardware. In Proceedings of the 7th annual neuro-inspired computational elements workshop (pp. 1-8).
>
> [2] Bezugam, S. S., Shaban, A., & Suri, M. (2022). Low power neuromorphic emg gesture classification. arXiv preprint arXiv:2206.02061.
>
> [3] Mohammadi, M., Chandarana, P., Seekings, J., Hendrix, S., & Zand, R. (2022). Static hand gesture recognition for American sign language using neuromorphic hardware. Neuromorphic Computing and Engineering, 2(4), 044005.
>
> [4] Chandarana, P., Mohammadi, M., Seekings, J., & Zand, R. (2022, October). Energy-efficient deployment of machine learning workloads on neuromorphic hardware. In 2022 IEEE 13th International Green and Sustainable Computing Conference (IGSC) (pp. 1-7). IEEE.

---

> ### Author Response · Authors · 2025-11-19
> **Response to Reviewer wECy (part 2)**
>
> This is a continued response to weakness 2.
>
> ***W2: There is a little lack of clarity between "incremental inference" and "early-exit": both concepts are related, yet they are distinct. The paper would benefit from a clearer delineation early on.***
>
> **ANSWER:** Early exit is one of the conditions for implementing incremental inference. In our terminology, incremental inference denotes the overall inference paradigm of SIREN: the network processes the event stream step by step, continuously updating its internal state and optionally refining its prediction as new event frames arrive, rather than committing to a fixed time window of length $T$. In contrast, early-exit refers to the decision mechanism we place on top of this incremental process: at each step we evaluate a confidence-based criterion (EMA + entropy + patience) to decide whether to stop and output the current prediction, or to continue integrating more events.
>
> We have clarified the terminology by explaining the relationship between "incremental inference" and "early-exit" in the revised manuscript. We now make this distinction explicit in Section 1 line 66 with the following text:
>
> *To realize the above incremental inference behavior, we require an inference exit mechanism, we implement a smooth exit mechanism with a patience parameter, enabling early-exit based on the entropy of confidence distribution.*

---

> ### Author Response · Authors · 2025-11-19
> **Response to Reviewer wECy (part 3)**
>
> This is a continued response to the question 1.
>
> ***Q1: Early-exit design choices need more justification.***
>
> We have incorporated the hyperparameter exploration results into Figure 6 on page 10 of the revised manuscript.
>
> ***(1) Explain how theta and kappa generalise across datasets. Are thresholds tuned per dataset? How sensitive are results to theta choice?***
>
> **ANSWER:** $\theta$ is the entropy threshold that decides whether we exit at a given step. When transferring to a new dataset, we first select a small validation split. If there is a target constraint on average exit time or energy, one can simply choose $\theta$ such that the resulting average exit step satisfies this requirement, since each $\theta$ corresponds to a specific average exit step. If there is no such constraint, one can instead compute the ACC–STEP Pareto front (as in Section 5.1, line 369) and pick a $\theta$ that best matches the desired accuracy–latency or loss-latency trade-off. On SL-Animals-DVS, we sweep 10 values of $\theta$ uniformly in $[0.001, 0.3]$ and obtain the ACC vs. STEP curves as shown in Fig. 6(d). We observe that performance is more sensitive to $\theta$ in the very early-exit regime, while sensitivity decreases once the average exit step becomes moderate.
>
> $\kappa$ controls how many consecutive “confident” steps are required before exiting, and is mainly introduced to avoid spurious early exits on realistic DVS sequences where different actions share similar prefixes (e.g., crouching vs. falling). On SL-Animals-DVS we sweep $\kappa \in \{1,\dots,5\}$ and report the accuracy–latency trade-off in Fig. 6(a). We observe that $\kappa=1$ already improves early-step accuracy over the naive max-probability baseline, while for $\kappa \ge 2$ a larger $\kappa$ makes the rule more conservative: the curves shift downward in the very early-exit regime but remain close once the average exit step becomes moderate. This suggests that $\kappa$ mainly controls how aggressive early exits are, rather than fundamentally changing the attainable accuracy, so we fix a moderate value of $\kappa$ across datasets without further tuning.
>
> ***(2) Show robustness: curves of accuracy vs. average step for different theta to show degradation.***
> **ANSWER:** Following the reviewer’s suggestion, we have added a robustness study for $\theta$ on SL-Animals-DVS. We sweep 10 values of $\theta$ uniformly in $[0.001, 0.3]$, and for each setting we vary the maximum allowed inference steps from $T=1$ to $T=16$ and plot the resulting ACC–STEP curves. As shown in Fig. 6(d), the curves for different $\theta$ almost lie on top of each other. The main effect of $\theta$ is to determine where along this common accuracy–latency Pareto curve the model is allowed to operate (i.e., how early it is permitted to exit), rather than to change the shape of the curve itself. This indicates that the early-exit rule is robust to the exact choice of $\theta$, which primarily serves to select an operating point on a stable ACC–STEP trade-off.
>
> ***(3) Clarify the role of EMA smoothing coefficient alpha -- any lag introduced and its effect on early exit latency.***
>
> **ANSWER:** At each step $t$, we maintain an exponentially smoothed confidence $\tilde c_t = \alpha \tilde c_{t-1} + (1-\alpha)c_t$, and apply the entropy–patience criterion on $\tilde c_t$ rather than on the raw $c_t$. Thus, $\alpha$ controls a trade-off between **stability** and **reactivity** of the exit decision: smaller $\alpha$ makes the rule react to single noisy frames, while larger $\alpha$ suppresses transient spikes. With this value, the EMA effectively averages over only the last few steps, so the additional lag it can introduce is at most a small, constant number of steps and is further bounded by the patience parameter $\kappa$ that already requires several consecutive confident steps. Empirically, varying $\alpha$ within a reasonable range mainly shifts the average exit step slightly while leaving the overall ACC–STEP trade-off almost unchanged; the early-exit behaviour is much more sensitive to the choice of $\theta$ than to $\alpha$ itself.
>
> We have added a clarification of the EMA coefficient $\alpha$ in Section 4.4 (around line 323) of the revised manuscript, as follows:
>
> Algorithm 1 details the inference process for an input event stream, where the event stream is split into $L$ event frames, and an EMA with coefficient $\alpha$ is used to smooth the confidence scores, suppressing transient spikes while introducing only a small temporal lag.

---

> ### Author Response · Authors · 2025-11-19
> **Response to Reviewer wECy (part 4)**
>
> This is a continued response to the question 2.
>
> ***Q2: The Causal Time Self-Attention (CTSA) softmax-free formulation is interesting. I recommend to give a short intuitive comparison to standard softmax attention (when does one outperform the other?), and discuss possible weaknesses.***
>
> **ANSWER:** We thank the reviewer for this suggestion. We have added, at the end of Appendix A.1.2, a short comparison between CTSA and standard softmax attention, together with a brief discussion of potential weaknesses. The added text reads:
>
> For reference, standard (causal) self-attention with softmax uses
> $$
> \alpha_{t,t'} ^m= \frac{\exp \big(\gamma\ Q_t K_{t'}^\top M_{t,t'} \big)}{\sum_{s \le t} \exp\big(\gamma\ Q_t K_{s}^\top M_{t,s}\big)}, \quad
> Attn_t ^m= \sum_{t' \le t} \alpha ^m_{t,t'} V_{t'} .
> $$
> with a temperature $\gamma>0$. Here the weights $\alpha^m_{t,t'}$ form a probability distribution over past steps $t' \le t$, which enforces strong competition: a few positions can dominate, and every new token changes the normalization over all previous ones.
>
> By contrast, CTSA replaces the softmax normalization with bounded scores $\beta_{t,t'}$ and length-normalized weights $w_{t,t'}$ (see Eq.~(A.3) in the appendix). Intuitively,
> $$
> CTSA_t = \Theta\Big(\sum_{t' \le t} w_{t,t'} V_{t'}\Big)
> $$
> preserves the sign and relative magnitude of the dot products $Q_t K_{t'}^\top$ and avoids global renormalization as $t$ grows. Together with the bounds above, this yields a streaming-friendly temporal operator whose forward and backward norms are controlled without softmax, and whose cost scales as $\mathcal{O}(r_Q r_K Z_t d_h)$ instead of $\mathcal{O}(Z_t d_h)$ for dense attention.
>
> In terms of when each is preferable, CTSA is particularly suitable in our setting of incremental, low-latency event processing: it naturally supports causal, stepwise updates, sparse spikes, and bounded temporal cost without recomputing a softmax over a growing window. Standard softmax attention may still be advantageous in large, offline transformers where sharp, highly selective attention maps are desired and energy/latency constraints are less critical, as its probability simplex can yield more peaked focus.
>
> A limitation of CTSA is that it does not produce a probability distribution over time and is more sensitive to the overall scale of $Q_t$ and $K_{t'}$: there is no implicit global normalization as in softmax. If feature scales are poorly controlled, CTSA can be less robust and may underperform softmax in purely accuracy-driven, high-capacity regimes. In SIREN we mitigate this with the per-head scaling $\sigma$, normalization of features (case (A) or (B) in Appendix A.1.2), and bounded effective history length $Z_t$, which empirically suffices on the DVS benchmarks considered in this work.

---

### Official Review · Reviewer_RFvo · 2025-11-02

**Soundness:** 3
**Presentation:** 2
**Contribution:** 2
**Rating:** 6
**Confidence:** 4

**Summary:**

This paper studies event-driven dynamic visual recognition with SNNs and proposes the Spiking Incremental Recognition Network, which performs incremental inference by updating states step-by-step and producing predictions once a confidence threshold is reached, rather than relying on fixed time windows. The design combines multiple existing components, diverse spiking neuron types, a spiking state-space model for multi-scale temporal modeling, causal time self-attention, and an early-exit mechanism, to better approximate ideal neuromorphic processing. Experiments on three DVS benchmarks show state-of-the-art accuracy while reducing inference steps and synaptic operations, indicating improved efficiency for potential edge deployment.

**Strengths:**

1. The paper tackles a relevant gap in SNNs by moving from fixed-window processing to incremental event-driven inference for neuromorphic deployment.

2. Experiments are comprehensive across multiple DVS datasets, with strong results on both accuracy and efficiency (steps/SOPs), supported by useful ablations.

3. Writing is clear and experiments are well-presented, with informative visualizations.

**Weaknesses:**

1. The approach largely integrates known mechanisms (multiple spiking neurons, S-SSM, causal attention, early exit). The conceptual novelty lies more in system integration rather than a fundamentally new algorithmic principle.

2. Related works on step-wise SNN inference and adaptive time-step SNN training frameworks need deeper positioning. For example: [a] Towards Low-Latency Event-Based Visual Recognition with Hybrid Step-Wise Distillation SNNs, [b] Adaptive Time-Step Training for Enhancing Spike-Based Neural Radiance Fields, and recent works on event-driven adaptive inference in neuromorphic computing

3. The main framework figure is visually weak and does not communicate the incremental inference mechanism clearly. A timeline or event-accumulation visualization could greatly help understanding.

4. Claims about neuromorphic deployment and continuous inference motivation are strong, but there is no hardware-based profiling or simulation beyond SOP estimates. Even lightweight profiling on Intel Loihi / NVIDIA DAVIS pipelines / microcontroller simulation would enhance significance.

**Questions:**

1. Is SIREN’s incremental strategy fundamentally dependent on S-SSM + CTSA, or could a simpler architecture yield similar gains?

2. Can the authors quantitatively isolate the contribution of each module (multi-neuron fusion, S-SSM, CTSA, early exit) under incremental inference?

3. How sensitive is performance to the confidence threshold? Is there a mechanism to guarantee monotonic confidence or prevent premature exit?

4. Is S-SSM implementable efficiently on Loihi-like accelerators?

5. Would SIREN extend to event-based tracking or segmentation? Does incremental inference scale to long sequences (e.g., E2Caltech)?

---

> ### Author Response · Authors · 2025-11-19
> **Response to Reviewer RFvo (part 1)**
>
> Thank you for taking the time to review. Please see our detailed response to the issues you have raised.
>
> ## Response to Weaknesses
>
> ***W1: The approach largely integrates known mechanisms (multiple spiking neurons, S-SSM, causal attention, early exit). The conceptual novelty lies more in system integration rather than a fundamentally new algorithmic principle.***
>
> **ANSWER:** We indeed leverage several well-established mechanisms, such as spiking neurons and early-exit mechanisms. However, we would like to clarify that our intended contribution is not to introduce a new gradient estimator or learning rule, but to provide a problem and system level formulation of step-adaptive incremental inference for SNNs.
>
> Conceptually, prior work on SNNs for event-driven vision predominantly assumes a fixed time step for both training and inference, while early-exit or dynamic inference has mostly been explored in ANN settings or at the frame/layer level. In contrast, we formulate event-driven recognition with SNNs as a sequential decision problem over timesteps, where the network must decide when it has accumulated sufficient evidence to stop, and we couple this decision with the internal spiking dynamics and state evolution. This step-adaptive inference view, to the best of our knowledge, has not been systematically studied for SNNs on real neuromorphic benchmarks.
>
> On the technical side, the individual components in SIREN are not inserted independently, but are designed to work together under the incremental inference constraint:
>
> - The spiking state-space module is instantiated so that its hidden state can be updated incrementally over a variable time horizon, remaining compatible with membrane potential dynamics and enabling efficient reuse of temporal context.
> - The causal self-attention is constrained to be strictly time-causal and incrementally updatable, which is crucial to preserve the semantics of early-exit decisions in an event-driven setting.
> - The early-exit rule is not a post-hoc heuristic attached to a fixed model; instead, it is integrated into the learning objective so that the network jointly optimizes accuracy and the time/energy budget under the proposed incremental regime.
>
> ***W2: Related works on step-wise SNN inference and adaptive time-step SNN training frameworks need deeper positioning. For example: [a] Towards Low-Latency Event-Based Visual Recognition with Hybrid Step-Wise Distillation SNNs, [b] Adaptive Time-Step Training for Enhancing Spike-Based Neural Radiance Fields, and recent works on event-driven adaptive inference in neuromorphic computing.***
>
> **ANSWER:** In the revised manuscript, we have expanded the Related Work section to better position step-wise SNN inference and adaptive time-step training.
> Specifically, we now include [a] Hybrid Step-Wise Distillation SNNs and [b] Adaptive Time-Step Training for Spike-Based NeRFs, together with recent neuromorphic adaptive-inference studies.
> These works have been incorporated into the “Dynamic Inference Strategies’’ and “Temporal Generalization in Training’’ paragraphs, respectively, with clearer distinctions from SIREN.
>
> The main framework figure is visually weak and does not communicate the incremental inference mechanism clearly. A timeline or event-accumulation visualization could greatly help understanding.
>
> ***W3: The main framework figure is visually weak and does not communicate the incremental inference mechanism clearly. A timeline or event-accumulation visualization could greatly help understanding.***
>
> **ANSWER:** In the revised manuscript, we have improved Figure 2 by adding explicit timelines for both the training and incremental inference procedures. We also expanded the caption to describe each module and the overall SIREN workflow in more detail. We believe the updated figure now conveys the incremental inference mechanism much more clearly.

---

> ### Author Response · Authors · 2025-11-19
> **Response to Reviewer RFvo (part 2)**
>
> This is a continued response to the weakness 4.
>
> ***W4: Claims about neuromorphic deployment and continuous inference motivation are strong, but there is no hardware-based profiling or simulation beyond SOP estimates. Even lightweight profiling on Intel Loihi / NVIDIA DAVIS pipelines / microcontroller simulation would enhance significance.***
>
> **ANSWER:** We currently do not have direct access to Loihi hardware. As far as we understand, INRC research members must first submit a project proposal and can only obtain access after it is approved. We have already initiated this application process; however, given the limited time available during the rebuttal period, it is unlikely that we can complete it in time. In the meantime, we have surveyed relevant literature, including both GPU-based simulations and evaluations on neuromorphic hardware. The comparison results are summarized below and indirectly demonstrate the deployment efficiency and performance of our approach.
>
> | Work                   | Task / Dataset                              | Baseline Platform                           | Neuromorphic Platform     | Main Efficiency Result                                                                                                                                                               |
> |------------------------|---------------------------------------------|---------------------------------------------|---------------------------|--------------------------------------------------------------------------------------------------------------------------------------------------------------------------------------|
> | Blouw et al. 2018$^{[1]}$      | Keyword spotting                            | CPU, GPU, Jetson, Movidius                  | Loihi                     | Loihi achieves the lowest energy cost per inference among all platforms at comparable accuracy.                                                                                      |
> | Bezugam et al. 2022$^{[2]}$    | EMG gesture classification                  | GPU                                         | Loihi (Nahuku-32)         | $\sim 983\times$ lower energy and $19\times$ lower latency than GPU (batch size = 50).                                                                                               |
> | Mohammadi et al. 2022$^{[3]}$  | Static ASL hand gestures                    | Intel NCS2                                  | Loihi                     | Up to $20.64\times$ lower power and $4.10\times$ lower energy than NCS2.                                                                                                            |
> | Chandarana et al. 2022$^{[4]}$ | Image classification (converted ANN to SNN) | Intel NCS2                                  | Loihi                     | Up to $27\times$ lower power and $5\times$ lower energy than NCS2.                                                                                                                  |
> | **Ours**       | DVS action recognition (SL-Animals-DVS)     | GPU (PyTorch, theoretical per-op model)     | no Loihi test currently   | $0.02$G SOPs and estimated $1.28$\,mJ for full 16-step inference; with average exit step $\sim 9.5$, the estimated energy is $\approx 0.76$mJ per sample, substantially lower than ANN baselines (36–45.75\,mJ for TORE-ResNet/GoogLeNet, 4.38\,mJ for EvT). |
>
> [1] Blouw, P., Choo, X., Hunsberger, E., & Eliasmith, C. (2019, March). Benchmarking keyword spotting efficiency on neuromorphic hardware. In Proceedings of the 7th annual neuro-inspired computational elements workshop (pp. 1-8).
>
> [2] Bezugam, S. S., Shaban, A., & Suri, M. (2022). Low power neuromorphic emg gesture classification. arXiv preprint arXiv:2206.02061.
>
> [3] Mohammadi, M., Chandarana, P., Seekings, J., Hendrix, S., & Zand, R. (2022). Static hand gesture recognition for American sign language using neuromorphic hardware. Neuromorphic Computing and Engineering, 2(4), 044005.
>
> [4] Chandarana, P., Mohammadi, M., Seekings, J., & Zand, R. (2022, October). Energy-efficient deployment of machine learning workloads on neuromorphic hardware. In 2022 IEEE 13th International Green and Sustainable Computing Conference (IGSC) (pp. 1-7). IEEE.

---

> ### Author Response · Authors · 2025-11-19
> **Response to Reviewer RFvo (part 3)**
>
> This is a continued response to questions 1-3.
>
> ***Q1: Is SIREN’s incremental strategy fundamentally dependent on S-SSM + CTSA, or could a simpler architecture yield similar gains?***
>
> **ANSWER:** SIREN’s incremental strategy is not fundamentally tied to the specific combination of S-SSM and CTSA. In our framework, the key driver of the efficiency–accuracy trade-off is theentropy-based early-exit gate with patience and EMA smoothing, which can in principle sit on top of different spiking backbones. That said, incremental inference is not universally applicable to all SNN architectures: if the network contains operations that explicitly couple multiple timesteps within a single forward pass (e.g., convolutions or pooling directly along the temporal dimension, which some SNNs use to aggregate temporal information), then it no longer supports strictly step-wise inputs and cannot be used with our stepwise incremental scheme.
>
> To verify that our gains do not solely rely on S-SSM + CTSA, we conducted additional ablations where we removed all attention modules and the S-SSM, leaving a purely convolutional SNN backbone with about 1.25M parameters and a fully connected classifier head. On SL-Animals-DVS, this simplified model achieves 77.33% accuracy. We also added a more comprehensive comparison against standard early-exit baselines, which is now reported in Section 5.3 and Figure 6.
>
> ***Q2: Can the authors quantitatively isolate the contribution of each module (multi-neuron fusion, S-SSM, CTSA, early exit) under incremental inference?***
>
> **ANSWER:** Yes. We have added quantitative ablations under incremental inference. Starting from the Conv SNN backbone, we progressively add SSA, S-SSM and CTSA, and for each variant we report accuracy–vs–average-exit-step curves. The results are presented in Sec. 5.3, Fig. 6(c).
>
> ***Q3: How sensitive is performance to the confidence threshold? Is there a mechanism to guarantee monotonic confidence or prevent premature exit?***
>
> **ANSWER:** In our implementation, the confidence threshold theta is tuned on a validation split. On SL-Animals-DVS, we sweep theta in the range [0, 0.3] and obtain a family of ACC–vs–average-step curves. We observe that SIREN is most sensitive to theta in the very early-exit regime (i.e., when the average exit step is very small): small changes in theta can noticeably shift the exit step. Once the average exit step moves into a moderate range (around 8–10 steps on SL-Animals-DVS), the curves become much smoother, and both the average step and accuracy vary only slightly when theta is perturbed. The results are presented in Sec. 5.3, Fig. 6(d).
>
> Regarding premature exit, our mechanism does not rely on a single-step confidence value. Instead, we combine (i) an exponential moving average (EMA) of the predicted probability vector and (ii) a patience rule: the model exits only if the smoothed confidence score exceeds theta for kappa consecutive steps and the predicted label remains unchanged. This EMA+patience scheme effectively suppresses short-term fluctuations in confidence and prevents unstable, premature exits, which is also reflected by the compact exit-step distributions reported in the paper.

---

> ### Author Response · Authors · 2025-11-19
> **Response to Reviewer RFvo (part 4)**
>
> This is a continued response to questions 4-5.
>
> ***Q4: Is S-SSM implementable efficiently on Loihi-like accelerators?***
>
> **ANSWER:** There already exist deployment methods for SSM-style architectures on neuromorphic hardware. Our contribution is algorithmic rather than a full chip design, but S-SSM is explicitly built from operations already supported by Loihi-like neuromorphic processors. At inference, all continuous-time parameters are pre-discretized offline, so S-SSM reduces to a per-step linear recurrence $x_{k+1} = \bar{N} x_k + \bar{S} u_k$, followed by a nonlinearity.
>
> Compared with standard (non-spiking) SSMs, S-SSM simply replaces the usual continuous-valued activation (e.g., ReLU) with a spike-generation mechanism (threshold-and-reset) and interprets $x_k$ as membrane state. This exactly matches the native computation of Loihi cores, which implement integer-weighted accumulations, programmable decay dynamics, local membrane state, and spike emission$^{[1]}$.
>
> Structurally, each S-SSM mode is a tiny real-valued recurrence (1D or a $2\times 2$ real block for complex modes) plus a local state register, i.e., a fixed-coefficient digital filter attached to a spiking neuron. Diagonal / block-diagonal state-space models with the same computational form have already been implemented efficiently on Loihi 2 (e.g., S4D) for streaming sequence processing, reporting orders-of-magnitude gains in energy and latency over GPU baselines$^{[2]}$. Sparse linear RNNs, when quantized and deployed on Loihi 2, further demonstrate large reductions in latency and energy compared to dense models on edge GPUs$^{[3]}$. Together with results showing that general dynamical systems can be compiled into spiking networks on Loihi-class hardware$^{[4]}$, these works indicate that modules with the same computational structure as S-SSM are not only implementable but can be realized efficiently on Loihi-like accelerators.
>
> [1] Davies, Mike, et al. "Loihi: A neuromorphic manycore processor with on-chip learning." Ieee Micro 38.1 (2018): 82-99.
> [2] Meyer, Svea Marie, et al. "A diagonal structured state space model on loihi 2 for efficient streaming sequence processing." 2025 Neuro Inspired Computational Elements (NICE). IEEE, 2025.
> [3] Pierro, Alessandro, et al. "Accelerating Linear Recurrent Neural Networks for the Edge with Unstructured Sparsity." arXiv preprint arXiv:2502.01330 (2025).
> [4] Voelker, Aaron Russell. Dynamical systems in spiking neuromorphic hardware. Diss. University of Waterloo, 2019.
>
>
> ***Q5: Would SIREN extend to event-based tracking or segmentation? Does incremental inference scale to long sequences (e.g., E2Caltech)?***
>
> **ANSWER:** ***W?: Would SIREN extend to event-based tracking or segmentation? Does incremental inference scale to long sequences (e.g., E2Caltech)?***
>
> **ANSWER:** We agree that extending SIREN beyond event-based classification is an important direction. Due to time and space constraints, we have not yet run dedicated experiments on tracking or segmentation, these remain future work.
>
> **Event-based tracking.** Conceptually, SIREN’s backbone (S-SSM + C-STSA / ChronoSpikFormer) is a generic spatiotemporal event encoder, so extending it to tracking is natural: one would retain the incremental encoder and replace the classification head with a lightweight detection / tracking head that predicts bounding boxes and identities over time. The incremental inference scheme would then update track states online and trigger more computation only when confidence drops, which fits the anytime, low-latency nature of tracking.
>
> **Event-based segmentation.** Extending SIREN to dense segmentation is more challenging, as it requires decoding temporally-encoded features into per-pixel masks and dealing with sparse, noisy event streams at high resolution. This would likely require a task-specific decoder (e.g., U-Net-style) and additional regularization for spatial consistency. While our backbone can in principle serve as the temporal encoder for such a system, a full segmentation design is beyond the scope of this work and we therefore leave it as a more substantial future extension.
>
> **Experiment on N-Caltech101** To show generalization beyond DVS Gesture and CIFAR10-DVS, we additionally evaluate ChronoSpikFormer on N-Caltech101. The results are reported in Appendix A.3.2 and summarized below:
>
> | Dataset        | Method            | Architecture           | Steps | Acc.(%) |
> |----------------|-------------------|------------------------|:-----:|:-------:|
> | **N-Caltech101** | QKFormer          | QKFormer-4-384         |  16   |  83.6   |
> |                | SpikingResformer  | SpikingResformer-Ti    |  16   |  81.3   |
> |                | Spikformer        | Spikformer-4-384       |  16   |  75.1   |
> |                | **Ours**          | ChronoSpikFormer-3-256 |  10   |  84.0   |

---

> > ### Comment · Reviewer_RFvo · 2025-11-27
> >
> > Thanks for the authors’ efforts in the rebuttal. While the paper has improved, it does not yet reach the level of a clear accept in my view, so I will keep my current score.

---

### Author Response · Authors · 2025-12-01
**Reviewer-Author Discussion Summary**

Dear Reviewers, AC, SAC, and PC,

We sincerely thank all reviewers for their time, effort, and constructive comments during the review and discussion phases, and we are grateful for the positive assessments of our work’s novelty in framing step-wise incremental inference to bridge the gap between fixed-step SNNs and event-driven neuromorphic operation (RFvo, wECy, g62k), its methodological design integrating S-SSM and C-STSA with a confidence-based early-exit mechanism (RFvo, wECy, g62k), its comprehensive experiments and competitive performance with reduced inference steps, SOPs, and energy consumption on multiple DVS benchmarks (RFvo, wECy, g62k, KzPq), and its clear presentation and informative visualizations (RFvo, wECy).

We also appreciate the reviewers’ follow-up comments during the discussion phase, which guided several important improvements to the manuscript, and we summarize the main changes below.

Following the suggestions of RFvo and KzPq, we revised the main framework figure and extended its caption to more clearly illustrate the incremental inference pipeline and architectural components (lines 197–204, page 4; Figure 2).

Following RFvo, wECy, and g62k, we toned down our claims regarding compatibility with neuromorphic hardware and explicitly clarified the limitations of our current energy analysis, as detailed in Response to Reviewer g62k’s Follow-up Comments (part 1) and reflected in the revised manuscript.

Following RFvo and wECy, we added experiments on the sensitivity of the early-exit behavior to the hyperparameters $\alpha$, $\kappa$, and $\theta$ as well as to architectural variants, and reported the resulting ACC–STEP trade-offs (Figure 6).

Following wECy, we added a comparison between CTSA and standard softmax attention, together with a discussion of potential weaknesses and suitable use cases (Appendix A.1.2).

Following g62k and KzPq, we added results on CIFAR10-DVS and N-Caltech101 to broaden the evaluation across standard DVS benchmarks (Appendix A.3.2, Table 8).

Following g62k, we extended the ablation study of SIREN’s architecture to more systematically isolate the contributions of SSA, S-SSM, and CTSA, covering all key module combinations (line 353, Table 1).

Following g62k, we added a comparison with a naive early-exit strategy on the same backbone (Figure 6(a),(b)).

Following g62k, we added a limitations section clarifying the spatial limitations of our model and the current absence of neuromorphic hardware deployment, which we leave for future work (Appendix A.4).

Following KzPq, we re-ran the TMC experiment at a uniform input resolution with corrected implementation details and updated the corresponding results (Table 2).

Following KzPq, we added a comparison with QKFormer on DVS128-Gesture (Table 2).

Following KzPq, we added a description of ChronoSpikFormer and clarified its relationship to SIREN (line 165).

Sincerely,

Authors

---

### Meta-Review · Area_Chair_2P5Y · 2026-01-06

**Summary:**

The paper proposes the SIREN architecture, which integrates a **S-SSM**, **C-STSA**, and **an entropy-based early-exit mechanism** to address the structural mismatch between fixed-step training in conventional SNNs and the event-driven nature of neuromorphic hardware. Reviewers generally acknowledge that the proposed incremental inference strategy effectively reduces the average number of inference steps and SOPs, and consider this aspect to be a meaningful advantage for low-latency edge deployment.

However, several important concerns remain, primarily regarding the level of methodological novelty and the consistency with core SNN principles. Some reviewers (notably Reviewer RFvo and Reviewer g62k) note that the proposed architecture largely consists of a combination of existing components. In addition, Reviewer g62k and Reviewer KzPq raise concerns that the inclusion of C-STSA and S-SSM, which involve real-valued multiplications, Softmax operations, and other non-local, non-spike-driven computations, departs from the sparse and event-driven computation paradigm that motivates spiking neural networks.

Overall, given the apparent “stacking of modules” nature of the proposed approach and the risk that continuous-valued components may weaken the intrinsic advantages of SNNs for hardware-efficient inference, the submission does not yet meet the standards of originality and conceptual coherence expected at ICLR. I therefore recommend rejection in its current form.

**Reviewer Concerns:**

Concerns addressed by the rebuttal:

- Clarity of the framework: Improved the presentation of the incremental inference framework by revising the main architecture figure and expanding its caption to better explain the inference process and architectural components.

- Ablation studies: Extended ablation experiments to systematically disentangle the contributions of key modules, including SSA, S-SSM, and CTSA.

- Experimental completeness: Strengthened experimental evaluation by adding results on additional standard DVS benchmarks, including CIFAR10-DVS and N-Caltech101.

- Implementation correctness: Corrected experimental settings and implementation details, including rerunning experiments under unified input resolutions.

Remaining concerns:

- Lack of evaluation on more challenging tasks and datasets: The authors do not provide additional experimental results on event tracking tasks or more complex datasets (e.g., large-scale or highly challenging benchmarks), which limits the assessment of the method’s generalizability.

- Absence of hardware-level validation: No experiments are conducted on neuromorphic or hardware platforms, which hinders the validation of the proposed method’s practical effectiveness and real-world applicability.

**Reviewer Scores:**

Reviewer Scores Based on rebuttal responses and explicit reviewer comments:

- Reviewer RFvo: 6 (Score maintained)

- Reviewer wECy: 8 (Concerns addressed; score maintained)

- Reviewer g62k: 4 (No substantial improvement in experimental depth or conclusions; score maintained)

- Reviewer KzPq: 2 (Experiments remain insufficient)

As the overall score is below the acceptance threshold, the paper is rejected.

---

### Decision · Program_Chairs · 2026-01-26

Reject